# IN-CONTEXT EDITING: LEARNING KNOWLEDGE FROM SELF-INDUCED DISTRIBUTIONS

**Siyuan Qi[1†,\*]  Bangcheng Yang[1\*]  Kailin Jiang[1,2\*]  Xiaobo Wang[1]**
**Jiaqi Li[1]    Yifan Zhong[1,3]    Yaodong Yang[1,3]    Zilong Zheng[1†]**
[1]State Key Laboratory of General Artificial Intelligence, BIGAI
[2]University of Science and Technology of China  [3]Peking University

## ABSTRACT

In scenarios where language models must incorporate new information efficiently without extensive retraining, traditional fine-tuning methods are prone to overfitting, degraded generalization, and unnatural language generation. To address these limitations, we introduce Consistent In-Context Editing (ICE), a novel approach leveraging the model's in-context learning capability to optimize toward a contextual distribution rather than a one-hot target. ICE introduces a simple yet effective optimization framework for the model to internalize new knowledge by aligning its output distributions with and without additional context. This method enhances the robustness and effectiveness of gradient-based tuning methods, preventing overfitting and preserving the model's integrity. We analyze ICE across four critical aspects of knowledge editing: accuracy, locality, generalization, and linguistic quality, demonstrating its advantages. Experimental results confirm the effectiveness of ICE and demonstrate its potential for continual editing, ensuring that the integrity of the model is preserved while updating information.

## 1 INTRODUCTION

In an ever-evolving world, it is crucial to update large language models (LLMs) to rectify outdated information and integrate new knowledge. Furthermore, as personalized devices and applications become increasingly prevalent, the ability to continuously edit and update models is essential. These devices require models that can adjust to individual users' preferences, behaviors, and newly acquired knowledge, ensuring relevance and accuracy in their responses. Updating large language models (LLMs) presents a significant challenge, as it often requires retraining from scratch—a process that is both computationally prohibitive and impractical. Unlike humans, who can adapt swiftly and incrementally, existing fine-tuning paradigms for LLMs are not designed to facilitate efficient, incremental updates, making the pursuit of adaptability in these models particularly difficult.

Knowledge editing [55] has emerged as a research area that addresses the challenge of efficiently updating LLM outputs in response to specific queries. It focuses on modifying particular pieces of knowledge in a language model $\mathcal{M}_\theta$ using query-response pairs $\{(\mathbf{q}_i, \mathbf{x}_i^*)\}_{i=1}^N$. For instance, given the query "*The president of the US is*", a model trained on outdated data might respond "*Joe Biden*", while the up-to-date response would be "*Donald Trump*".

This is typically achieved by maximizing the probability $p_\theta(\mathbf{x}^*|\mathbf{q})$ using fine-tuning. However, this approach can be brittle in knowledge editing scenarios, where incorporation of new information with **minimal data** is crucial [28]. This is because fine-tuning often minimizes the distance (e.g., cross-entropy) between predictions and **one-hot target distributions** $\delta_{\mathbf{x}^*}(x)$, which can cause overfitting and result in model degradation or even model collapse, especially when data is scarce.

Various strategies have been proposed to address this problem, including constraining the gradient or weights [59; 28] and adopting parameter-efficient fine-tuning approaches [54]. However, these methods still rely on one-hot target distributions, failing to fully mitigate overfitting.

---

† Corresponding author. * Equal contribution.

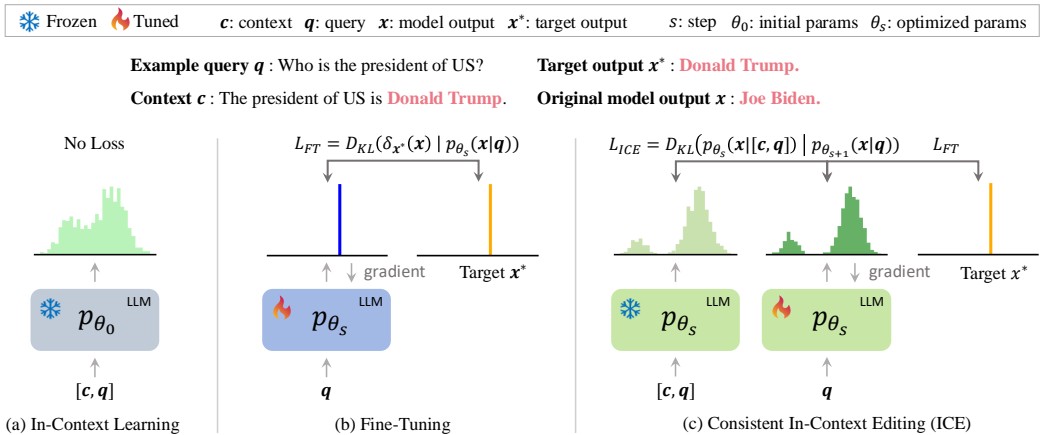

Figure 1: Overview. (a) **In-Context Learning**: Utilizes context prompts without modifying model parameters, allowing dynamic adaptation but lacking parameter updates. (b) **Traditional Fine-Tuning**: Minimizes the distance between predictions and a one-hot target ($\delta_{x^*}$) using cross-entropy loss ($L_{ft}$), often leading to overfitting. (c) **Consistent In-Context Editing (ICE)**: Adds a contextual loss ($L_{ice}$) to the traditional fine-tuning loss ($L_{ft}$). $L_{ice}$ minimizes the divergence between model outputs with and without a context prompt, aligning the model toward internalizing new knowledge. This helps ICE achieve effective knowledge incorporation while preserving general model stability.

To address these limitations, we introduce Consistent In-Context Editing (ICE), a novel method that learns toward a contextual distribution to effectively internalize new knowledge. Specifically, ICE guides the model's output distribution $p_\theta(\mathbf{x}|\mathbf{q})$ to align with a contextual distribution $p_\theta(\mathbf{x}|[\mathbf{c}, \mathbf{q}])$ induced by a context $\mathbf{c}$ that includes the target knowledge. We minimize the KL divergence between these two distributions, encouraging the model to internalize the new knowledge.

However, the initial contextual distribution may not always perfectly align with the desired update target, so we dynamically adjust it during optimization. We achieve this by combining the contextual loss as a regularization term with the original fine-tuning loss. As the fine-tuning loss is minimized, both the output distribution and the contextual output distribution are guided toward the desired target. The contextual loss, serving as a regularization term, confines the extent of these modifications, thereby ensuring the model's integrity and preventing unintended degradation. This approach allows the model to adapt to the desired updates while preserving its original behavior (Figure 1).

While there exist methods that utilize in-context learning for knowledge editing [57], they add new information to context prompts without modifying the model parameters, requiring the models to always operate with the updated context. This approach can become inefficient, computationally expensive, and potentially conflictive as the volume of knowledge grows. In contrast, ICE directly updates the model parameters, allowing it to manage a growing and evolving knowledge base.

Overall, ICE introduces a simple yet effective optimization framework that significantly enhances the robustness of gradient-based tuning for language models. At each optimization step, ICE samples in-context sequences and minimizes the difference between outputs with and without context, alongside the fine-tuning loss. This process ensures accurate incorporation of new knowledge, prevents overfitting and preserves the integrity of previously learned information.

We conduct extensive experiments on four datasets, obtaining promising results across four key dimensions: accuracy, locality, generalization, and linguistic quality. In addition, we evaluate ICE 's performance without any adjustments in continual editing scenarios, where the model undergoes sequential updates, each time with a piece of new knowledge. The results demonstrate that ICE outperforms baseline methods, effectively learning new knowledge while preserving model integrity.

The primary contributions of this paper are: 1) We introduce in-context editing (ICE), a novel knowledge editing approach that learns toward a contextual distribution rather than a one-hot target, offering a more robust alternative to traditional fine-tuning. 2) We develop an optimization framework that refines the target contextual distribution using a gradient-based algorithm, enabling dynamic adaptation of the model to correctly incorporate new knowledge. 3) We provide empirical evidence

demonstrating the effectiveness of ICE, showcasing its potential for continual editing by seamlessly integrating new information while preserving the integrity of existing knowledge.

## 2 PRELIMINARIES AND RELATED WORK

### 2.1 KNOWLEDGE EDITING: PROBLEM SETUP

The objective of knowledge editing is to incorporate new facts into a language model $\mathcal{M}_\theta$ through query-response pairs $\{(\mathbf{q}_i, \mathbf{x}_i^*)\}_{i \in [1,N]}$. Here $\mathbf{q}$ is the query that triggers the retrieval of factual knowledge from $\mathcal{M}_\theta$, such as "*The president of the US is*", with $\mathbf{x}^*$ being the intended response after editing, *e.g.*, "*Donald Trump*". This integration is typically done by maximizing the probability $p_\theta(\mathbf{x}^*|\mathbf{q})$. Conventionally, a knowledge editing algorithm is assessed across four key dimensions.

**Edit Success** measures the ability of the model to produce the edited response $\mathbf{x}^*$ for a query $\mathbf{q}$. Let $\mathcal{D}_e$ represent the query-response pairs and $\mathbf{1}[\cdot]$ be the indicator function, the metric is defined as:

$$\text{succ} = \mathbb{E}_{(\mathbf{q},\mathbf{x}^*) \sim \mathcal{D}_e}[\mathbf{1}[\arg\max_{\mathbf{x}} p_\theta(\mathbf{x}|\mathbf{q}) = \mathbf{x}^*]].$$

**Portability** assesses how well the model generalizes the knowledge for rephrased or logically related queries within the edit scope $\mathcal{D}_q$. For the aforementioned example, a related query could be "*The first lady of the US is*", with the target being "*Melania Trump*" instead of "*Jill Biden*".

$$\text{port} = \mathbb{E}_{(\mathbf{q},\mathbf{x}^*) \sim \mathcal{D}_q \setminus \mathcal{D}_e}[\mathbf{1}[\arg\max_{\mathbf{x}} p_\theta(\mathbf{x}|\mathbf{q}) = \mathbf{x}^*]].$$

**Locality** evaluates if the model maintains original predictions for queries outside the edit scope:

$$\text{loc} = \mathbb{E}_{(\mathbf{q},\mathbf{x}^*) \sim \mathcal{D} \setminus \mathcal{D}_q}[\mathbf{1}[\arg\max_{\mathbf{x}} p_\theta(\mathbf{x}|\mathbf{q}) = \mathbf{x}^*]].$$

**Fluency** estimates the linguistic quality of the post-edit model's output [56], given by a weighted sum of bi- and tri-gram entropies, given $f_n$ as the n-gram distribution:

$$\text{flu} = -\sum_{n=2}^{3} w_n \sum_{\mathbf{x}} f_n(\mathbf{x}) \log f_n(\mathbf{x}).$$

### 2.2 KNOWLEDGE-EDITING APPROACHES

**Weight Frozen** The first family of methods for knowledge editing keeps the original model frozen while leveraging external tools. Techniques proposed by [31; 33; 27; 35; 58; 49; 48; 22; 4; 40; 47] enhance the model's adaptability to new knowledge using external memory. Other approaches inject additional parameters into the model to incorporate new knowledge [9; 19; 14; 36; 54]. Additionally, some methods attempt to embed the knowledge directly into prompts to generate post-edit responses, utilizing the model's in-context learning abilities [57; 6; 24]. However, these methods result in an ever-growing memory/model, which can become problematic over time as knowledge accumulates.

**Weight Modified** Another line of work, which our method focuses on, involves editing the model's weights to integrate new knowledge. Approaches include direct fine-tuning [59; 23; 34], meta-learning-driven approaches [43; 8; 30; 41], and targeted network editing [10; 7; 25; 5; 38; 52; 44; 20; 13; 15; 16; 1; 11; 53; 51; 26; 12; 46; 39; 37]. These methods aim to incorporate target knowledge and employ various techniques to ensure the locality of the edits. Techniques include constraining the gradient of parameters [59], adopting parameter-efficient approaches [54], and applying statistical constraints on the weights, with notable examples such as ROME [28] and MEMIT [29]. In this paper, we propose tuning the model towards a self-generated distribution instead of a one-hot target.

**In-Context Learning** In-context learning refers to the ability of language models to use the information provided in the input context to perform tasks without altering the model's parameters [2]. Previous research has applied contextual knowledge by prompting the model [57; 6]. To enhance models leveraging in-context learning, various strategies have been explored, such as distilling contextual knowledge [42; 18] and compressing the context into a gist token [32]. However, [42] necessitates maintaining two copies of the model weights, while [32] requires the addition of an extra token to facilitate the injection of new knowledge. There are also methods tuning the model through

meta-learning techniques [3], and examining the consistency between context and knowledge [24; 17]. However, these approaches do not modify the model weights. In contrast, our method introduces a novel approach to utilize in-context learning by creating a learning target and framework for model editing, thereby providing an innovative way to integrate contextual information into the model's knowledge base.

# 3 METHODOLOGY: LEARNING KNOWLEDGE FROM CONTEXTS

We consider an auto-regressive generative language model $p_\theta$ parameterized by $\theta$, where $p_\theta(\mathbf{x}) = p_\theta(x_{1:T})$ denotes the probability of a sequence $\mathbf{x} = x_{1:T}$. The model factorizes the sequence into individual tokens $x_t$ and models the probability auto-regressively: $p_\theta(x_{1:T}) = p_\theta(x_1) \prod_{t=1}^{T-1} p_\theta(x_{t+1}|x_{1:t})$. Given new knowledge as a query-answer pair $(\mathbf{q}, \mathbf{x}^*)$, our primary goal is to update the model parameters $\theta$ to maximize $p_\theta(\mathbf{x}^*|\mathbf{q})$ while keeping the model's responses to unrelated queries unchanged.

**Running Example:** To illustrate our approach, suppose we want to update a language model to reflect that the current president of the United States is *Donald Trump*, whereas it currently outputs *Joe Biden* when queried with "*The president of the US is*". The model needs to be updated so that it outputs *Donald Trump* for this query, without affecting its performance on unrelated queries.

## 3.1 VANILLA FINE-TUNING

A straightforward approach to editing a model's knowledge is fine-tuning, which involves minimizing the cross-entropy loss between the model's predictions and the target knowledge. This is equivalent to minimizing the Kullback-Leibler (KL) divergence between the **one-hot target distribution** $\delta_{\mathbf{x}^*}(\mathbf{x})$ and the model's predicted distribution:

$$\mathcal{L}_{\text{FT}} = D_{\text{KL}}(\delta_{\mathbf{x}^*}(\mathbf{x}) \,||\, p_\theta(\mathbf{x}|\mathbf{q})). \tag{1}$$

In our running example, this means we fine-tune the model to assign maximum probability to the sequence "*Donald Trump*" given the query "*The president of the US is*". While this approach can effectively update the model's response for a specific query, it has significant drawbacks. The use of a one-hot target distribution often leads to overfitting, causing the model to degrade, suffer from catastrophic forgetting, or even collapse, resulting in unnatural or repetitive outputs.

## 3.2 FINE-TUNING WITH SAMPLING

To try to address the above issues, we can use diverse and representative data distributions during fine-tuning to enhance the model's adaptability and generalization. A potential strategy is to employ a softer distribution generated by the model itself in a bootstrapping manner, iteratively enhancing its performance. Unlike the hard one-hot distribution, this approach involves fine-tuning the model using its own sampled sequences, conditioned on the target $\mathbf{x}^*$ of length $m$. Specifically, we can set the concatenation of each query and target $[\mathbf{q}, \mathbf{x}^*]$ as the input, sample multiple sequences from the model itself, and use them as the fine-tuning targets:

$$\mathcal{L}_{\text{FT}}^* = D_{\text{KL}}(\delta_{\mathbf{x}^*}(x_{1:m})p_\theta(x_{>m}|[\mathbf{q}, \mathbf{x}^*]) \,||\, p_\theta(\mathbf{x}|\mathbf{q})). \tag{2}$$

However, as we show in Observation 1, this approach does not alleviate the overfitting problem and is effectively equivalent to the vanilla fine-tuning method.

**Observation 1.** *The objective of fine-tuning with samples is equivalent to the objective of traditional fine-tuning, i.e., $\mathcal{L}_{FT}^* = \mathcal{L}_{FT}$ (see § A.2 for a proof).*

This implies that the model cannot learn and improve on its own **without external inputs**, highlighting the necessity for our method, which will be introduced in the following sections.

## 3.3 IN-CONTEXT TUNING WITH SAMPLING

To address the ineffectiveness of the naive sampling approach in § 3.2, we introduce *extra information that guides the model towards a new distribution that aligns with the target, while maintaining similarity to its original distribution*. Specifically, we leverage the in-context learning capabilities of

language models by prepending context prompts $\mathbf{c}$ to the queries $\mathbf{q}$, where $\mathbf{c}$ is the new knowledge to be learned. For our example, we can create a context such as "*Donald Trump is the current president of the United States.*", and prepend this context to the query. This induces a new contextual distribution $p_{\theta_0}(\mathbf{x}|[\mathbf{c}, \mathbf{q}])$ that incorporates the desired knowledge through the context, while keeping minimal changes to the model. We can define the loss function as:

$$\mathcal{L}_{\text{sample}} = D_{\text{KL}}(p_{\theta_0}(\mathbf{x}|[\mathbf{c}, \mathbf{q}]) \,||\, p_\theta(\mathbf{x}|\mathbf{q})), \tag{3}$$

where $\theta_0$ are the initial parameters of the model, and $p_\theta(\mathbf{x}|\mathbf{q})$ is the updated model's distribution. In this formulation, there is no explicit target sequence $\mathbf{x}^*$; instead, the desired information is implicitly conveyed through the context $\mathbf{c}$. The effectiveness of this method relies on the relevance of the context and the model's ability to utilize it effectively.

### 3.4 CONSISTENT IN-CONTEXT EDITING (ICE)

While the loss $\mathcal{L}_{\text{sample}}$ introduces context, it does not guarantee that the model will produce accurate responses, as the initial distribution $p_{\theta_0}(\mathbf{x}|[\mathbf{c}, \mathbf{q}])$ may not reflect the correct target due to limitations in the model's ability to follow the context. Therefore, we propose to refine the target contextual distribution in a way that ensures the model internalizes the new knowledge.

We introduce a consistency condition that the updated model parameters $\theta$ should satisfy:

$$p_\theta(\mathbf{x}|[\mathbf{c}, \mathbf{q}]) = p_\theta(\mathbf{x}|\mathbf{q}). \tag{4}$$

This condition implies that, after updating, the model's predictions should be the same whether or not the context $\mathbf{c}$ is provided, indicating that the knowledge from $\mathbf{c}$ has been internalized. To enforce this, we define the *in-context editing loss* $\mathcal{L}_{\text{ICE}}$ as:

$$\mathcal{L}_{\text{ICE}} = D_{\text{KL}}(p_\theta(\mathbf{x}|[\mathbf{c}, \mathbf{q}]) \,||\, p_\theta(\mathbf{x}|\mathbf{q})). \tag{5}$$

To ensure the model produces the correct target sequence $\mathbf{x}^*$, we also include the fine-tuning loss:

$$\mathcal{L} = \mathcal{L}_{\text{FT}} + \lambda \mathcal{L}_{\text{ICE}}, \tag{6}$$

where $\lambda$ is a hyperparameter balancing the two loss terms.

**Optimizing $\mathcal{L}_{\text{ICE}}$:** The in-context editing loss $\mathcal{L}_{\text{ICE}}$ involves two distributions that depend on the model parameters $\theta$, making direct optimization challenging. Directly propagating the loss through both distributions is not desirable, as we aim for a uni-directional optimization: we do not intend to draw $p_\theta(\mathbf{x}|[\mathbf{c}, \mathbf{q}])$ towards $p_\theta(\mathbf{x}|\mathbf{q})$. To address this, we adopt an iterative, gradient-based approach. At each optimization step $s$, we treat $p_{\theta_s}(\mathbf{x}|[\mathbf{c}, \mathbf{q}])$ as a fixed target distribution (using the current parameters $\theta_s$) and update the model parameters to minimize the divergence to $p_{\theta_{s+1}}(\mathbf{x}|\mathbf{q})$. This process is formalized as:

$$\theta_{s+1}^* = \underset{\theta_{s+1}}{\arg\min} \, \mathcal{L}_{\text{ICE}}^{(s)} = \underset{\theta_{s+1}}{\arg\min} \, D_{\text{KL}}(p_{\theta_s}(\mathbf{x}|[\mathbf{c}, \mathbf{q}]) \,||\, p_{\theta_{s+1}}(\mathbf{x}|\mathbf{q})). \tag{7}$$

By iteratively updating $\theta$, we ensure that the model's predictions with and without the context converge, satisfying the consistency condition in Equation 4.

**Optimizing the Combined Loss $\mathcal{L}$:** To optimize the total loss as defined in Equation 6, we sample sequences $\mathbf{x_c}$ from the model conditioned on $[\mathbf{c}, \mathbf{q}, \mathbf{x}^*]$ and maximize the likelihood of the combined sequence $[\mathbf{x}^*, \mathbf{x_c}]$. This process is equivalent to optimizing the combined loss $\mathcal{L}$. If the sampling is not conditioned on the target, we would be solely optimizing $L_{\text{ICE}}$. This approach is algorithmically convenient, and we provide a proof of this equivalence in § A.3. To prevent the model from drifting too far from the initial parameters (thus preserving unrelated knowledge), we employ gradient clipping techniques inspired by constrained fine-tuning methods [59]. The detailed algorithm is presented in Algorithm 1.

**Context Generation:** The context $\mathbf{c}$ can be generated automatically by extracting or synthesizing relevant information related to the target knowledge. In practice, this can be achieved using language models or APIs to generate summaries or statements that convey the new information. In our experiments, we used GPT-4 to create effective contexts (details provided in § C).

## 3.5 DISCUSSION

Our method aims to achieve several objectives simultaneously. The **accuracy** of our method is ensured by the fine-tuning loss $\mathcal{L}_{\text{FT}}$, which requires the model produces the correct target output $\mathbf{x}^*$ for the query $\mathbf{q}$. The **linguistic quality** is maintained by the in-context editing loss $\mathcal{L}_{\text{ICE}}$, which encourages the model to align its output distribution with a broader, context-induced distribution, helping prevent overfitting and maintaining the naturalness and diversity of the generated text.

To understand how our method maintains **locality** (i.e., minimal impact on unrelated queries) and promotes **generalization**, we consider all possible query-response pairs $(\mathbf{q}, \mathbf{x})$ and partition them into two sets: those related to the target knowledge ($\mathcal{D}_q$) and those unrelated ($\mathcal{D}_{\neg q}$). The in-context editing loss can be decomposed as:

$$\begin{aligned}
\mathcal{L}_{\text{ICE}} &= D_{\text{KL}}(p_\theta(\mathbf{x}|[\mathbf{c}, \mathbf{q}]) \,||\, p_\theta(\mathbf{x}|\mathbf{q})) \\
&= \sum_{(\mathbf{q}, \mathbf{x}) \sim \mathcal{D}_q \cup \mathcal{D}_{\neg q}} p_\theta(\mathbf{x}|[\mathbf{c}, \mathbf{q}]) \log\left(\frac{p_\theta(\mathbf{x}|[\mathbf{c}, \mathbf{q}])}{p_\theta(\mathbf{x}|\mathbf{q})}\right).
\end{aligned} \tag{8}$$

For queries unrelated to the target knowledge (i.e., $\mathbf{q} \in \mathcal{D}_{\neg q}$), the context $\mathbf{c}$ should have minimal effect, so the loss encourages the model to keep its original responses, ensuring locality. For related queries ($\mathbf{q} \in \mathcal{D}_q$), the loss promotes generalization, as the model learns to apply the new knowledge to various relevant contexts. Thus the effectiveness of our method relies on several assumptions:

*The context is related to the knowledge.* The context provided in the prompts must be pertinent and relevant to the knowledge needed for generating accurate and coherent responses. This relevance ensures that the additional information introduced through the context is meaningful and enhances the model's understanding of the query.

*The model attends to the context.* The model must be capable of attending to and incorporating the contextual information provided in the prompts. During the fine-tuning process, the model effectively uses the context as part of its input, influencing its predictions and overall performance.

*The model generalizes from the context to related knowledge.* Given the relevant context, the model should be able to generalize from the specific information in the context to broader or related knowledge areas. This generalization enables the model to generate responses that are not only contextually coherent but also enriched with additional details inferred from the context. Techniques like chain-of-thought [50] can potentially be employed in the context prompt to enhance the model's generalization capability.

The computational demands of our pipeline can be heavier than vanilla fine-tuning, as it involves multiple sampling steps and depends on GPT-4 for context generation. However, the computational burden may not be as substantial as it appears: 1) Since sampling only necessitates a forward pass of the model, the computational cost is significantly lower than that of training the model. 2) We are considering scenarios with very limited training data, as is the case in the knowledge editing task.

## 4 EXPERIMENTS

### 4.1 EXPERIMENT SETTINGS

**Datasets and Model** We evaluate the performance of ICE with four datasets from KnowEdit [55], which are commonly used for knowledge insertion and modification. Detailed statistics on the selected datasets can be seen in Table 1. Among the datasets, the WikiBio dataset does not include related hopping question data necessary for evaluating the portability metric. To ensure a fair comparison, we use Llama2-7b-chat, which is the same model as used in the original survey [55].

**Metrics** We use the metrics from § 2.1 but note one limitation in not penalizing semantically meaningless sentences or repetitive long patterns (§ D.1). Hence we add perplexity as an additional measure, which measures how well the pre-trained model predicts the generated outputs from the fine-tuned models. Assuming the original model is well-trained, the perplexity score reflects the language quality of the fine-tuned model and how far it has drifted. In our case, perplexity can also increase due to the novelty of edited knowledge, so we introduce a normalized perplexity ratio $\text{PPL}_r$

Table 1: Statistics on the evaluation datasets.

| | Knowledge Insertion | Knowledge Modification | | |
|---|---|---|---|---|
| | $\text{WikiData}_{recent}$ | ZsRE | WikiBio | $\text{WikiData}_{counterfact}$ |
| Type | Fact | QA | Hallucination | Counterfact |
| Train | 570 | 10,000 | 592 | 1,455 |
| Test | 1,266 | 1230 | 1,392 | 885 |

Table 2: Main results on knowledge insertion and question-answering datasets of Llama2-7b-chat.

| | $\text{WikiData}_{recent}$ | | | | | ZsRE | | | | |
|---|---|---|---|---|---|---|---|---|---|---|
| | Edit Succ. ↑ | Portability ↑ | Locality ↑ | Fluency ↑ | $\text{PPL}_r$ ↓ | Edit Succ. ↑ | Portability ↑ | Locality ↑ | Fluency ↑ | $\text{PPL}_r$ ↓ |
| ROME | 97.25 | 36.58 | 30.40 | 581.00 | 107.47 | 96.66 | 52.90 | 26.61 | 573.02 | 53.88 |
| MEMIT | 97.03 | 37.00 | 29.28 | 573.06 | 87.17 | 95.61 | 52.73 | 24.79 | 563.42 | 38.67 |
| FT-L | 45.63 | 34.73 | 34.80 | 558.91 | 68.92 | 43.60 | 43.90 | 51.38 | 560.94 | 30.36 |
| FT-M | 100.00 | 59.28 | 41.54 | 587.17 | 70.64 | 100.00 | 54.47 | 53.84 | 580.10 | 27.33 |
| ICE | 100.00 | 61.02 | 46.39 | 585.58 | 34.08 | 100.00 | 55.52 | 56.97 | 562.70 | 15.50 |

to address this (§ D.1). The ratio compares the perplexity of the generation post-target token to that of the combined prompt and target token.

**Methods** We use 4 representative tuning methods for comprehensive comparisons. ROME [28] and MEMIT [29] employ a causal method to locate and edit only the related parameters to improve the locality. The other two methods FT-L [28] and FT-M [55] fine-tunes specific layers of the feed-forward network to maximize the probability of all tokens in the target sequence. In the survey [55], the FT-M model demonstrated nearly the best performance.

**Implementation details** The contexts **c** are given by GPT-4 by summarizing the target knowledge. For layer updates, ROME updates one layer for GPT2 with layer 17 and Llama2 with layer 5. For both ICE and other baselines (FT-M and FT-L), five layers are updated following MEMIT [29], for GPT2 with layers 13,14,15,16,17 and Llama2 with layers 4,5,6,7,8. Results of GPT2 can be found at § D.3. We follow the usage of other parameters in ROME and MEMIT which have been found to provide the best performance. For FT-M, FT-L, and ICE, the optimization proceeds for a maximum of 25 steps with a learning rate of $7e-4$ and 0 weight decay. For all results except the ablation study, we used $\lambda = 1.0$ for ICE without deliberate tuning.

## 4.2 MAIN RESULTS

Table 2 and Table 3 show the main performance metrics of ICE. Notably, the FT-M method remains the strongest baseline, as corroborated by the findings in [55]. As seen in the results, ICE demonstrates outstanding performance on the measures.

**Accuracy** ICE consistently achieves nearly perfect edit accuracy across all datasets, outperforming most baselines and matching the performance of the strongest baseline FT-M.

**Locality and portability** As accuracy increases, the locality tends to decrease due to the inherent perturbations introduced. Furthermore, there tends to be an inverse relationship between model locality and portability; locality implies minimal model changes, whereas portability necessitates the model's ability to generalize to related knowledge. Despite this trend, ICE not only achieves a near-perfect accuracy comparable to FT-M but also consistently outperforms baseline methods in terms of locality and portability, aligning with the analysis presented in § 3.5. While matching the near-perfect accuracy with FT-M, ICE demonstrates consistently better locality and portability than the baseline methods, matching our expectation discussed in § 3.5. Compared to ROME, MEMIT, and FT-T, ICE shows approximately 30% higher portability on the $\text{WikiData}_{counterfact}$ and $\text{WikiData}_{recent}$ datasets. This discrepancy highlights that by leveraging in-context learning to adapt to a contextual distribution, ICE achieves better generalization. Additionally, ICE performs over 15% better in terms of locality on both datasets, preserving unrelated knowledge by enhancing the robustness of gradient-based tuning. A minor performance degradation of 99.88% is observed on

Table 3: Main results on knowledge modification datasets of Llama2-7b-chat.

| | WikiBio | | | | WikiData$_{counterfact}$ | | | | |
|---|---|---|---|---|---|---|---|---|---|
| | Edit Succ. ↑ | Locality ↑ | Fluency ↑ | PPL$_r$ ↓ | Edit Succ. ↑ | Portability ↑ | Locality ↑ | Fluency ↑ | PPL$_r$ ↓ |
| ROME | 95.83 | 68.38 | 617.67 | 3.70 | 98.68 | 42.45 | 21.13 | 585.40 | 109.97 |
| MEMIT | 94.54 | 69.96 | 616.65 | 3.51 | 98.13 | 44.16 | 19.48 | 576.26 | 122.48 |
| FT-L | 59.41 | 28.94 | 615.50 | **1.89** | 36.13 | 29.37 | 38.37 | 566.55 | 89.24 |
| FT-M | **100.00** | 35.34 | **618.12** | 3.67 | **100.00** | 72.39 | 40.76 | **586.80** | 54.71 |
| ICE | 99.88 | **70.60** | 617.88 | 2.15 | **100.00** | **73.49** | **45.88** | 583.29 | **18.95** |

the WikiBio dataset. This could be attributed to the diversity across datasets, which can introduce slight variations in performance within an acceptable margin.

**Fluency and PPL$_r$**   To evaluate the linguistic quality, we computed fluency and perplexity. ICE demonstrates reasonably good fluency, frequently ranking among the top performers. While other methods might show slightly higher fluency in single edits, ICE achieves significantly higher fluency in the continual editing case (§ 4.4). Moreover, ICE consistently exhibits lower perplexity, signaling better and more natural language model performance. It maintains robust performance across all metrics when editing new knowledge while preserving the integrity of existing information.

Table 4: Ablation results. The second row is the closest to fine-tuning (§ 3.1 and § 3.2).

| Dynamic | Context | | ZsRE | | | | | WikiData$_{counterfact}$ | | | |
|---|---|---|---|---|---|---|---|---|---|---|---|
| | | Edit succ.↑ | Portability ↑ | Locality ↑ | Fluency ↑ | PPL$_r$ ↓ | Edit succ. ↑ | Portability ↑ | Locality ↑ | Fluency↑ | PPL$_r$ ↓ |
| ✓ | ✓ | **100.00** | **55.52** | 56.97 | 562.70 | 15.50 | **100.00** | **73.49** | 45.88 | 583.29 | **8.92** |
| ✓ | ✗ | 99.60 | 45.95 | 55.40 | 544.55 | **6.90** | 99.66 | 67.34 | 44.42 | 568.98 | 12.50 |
| ✗ | ✓ | 99.94 | 53.27 | 62.90 | 573.97 | 26.39 | **100.00** | 70.14 | 50.05 | **589.97** | 31.71 |
| ✗ | ✗ | 99.94 | 53.84 | **65.64** | **578.97** | 25.71 | 99.93 | 69.93 | **55.12** | 589.04 | 35.70 |

Table 5: Ablation results for different values of λ.

| λ | | ZsRE | | | | | WikiData$_{recent}$ | | | |
|---|---|---|---|---|---|---|---|---|---|---|
| | Edit succ.↑ | Portability ↑ | Locality ↑ | Fluency ↑ | PPL$_r$ ↓ | Edit succ. ↑ | Portability ↑ | Locality ↑ | Fluency↑ | PPL$_r$ ↓ |
| 0.6 | 99.71 | 50.65 | **59.54** | **584.84** | 561.29 | 99.93 | 58.28 | 46.93 | 589.77 | 179.4 |
| 0.8 | 99.81 | 51.59 | 58.82 | 582.91 | 281.14 | 99.95 | 59.12 | 47.36 | 591.84 | 142.03 |
| 1.0 | **100.00** | **55.52** | 56.97 | 562.70 | **15.50** | **100.00** | **61.02** | 46.39 | 585.58 | **34.08** |
| 1.2 | 99.87 | 52.08 | 58.54 | 581.04 | 324.47 | 99.98 | 59.69 | **47.51** | 589.49 | 280.32 |
| 1.4 | 99.90 | 51.91 | 58.18 | 584.17 | 287.25 | **100.00** | 59.96 | 46.53 | **591.98** | 154.77 |

## 4.3 ABLATION STUDIES

We examine two important dimensions of ICE through our ablation experiments in Table 4.

**Firstly**, we analyze the impact of using a dynamic training target. Specifically, we investigate whether sequences are generated from the original model throughout training or from a modified model. In other words, in the first variant of our algorithm, the target distribution $p_{\theta_s}(\mathbf{x}|[\mathbf{c}, \mathbf{q}])$ in Equation 7 remains static during optimization, meaning the weight of the with-context target distribution does not change, i.e., $p_{\theta_s}(\mathbf{x}|[\mathbf{c}, \mathbf{q}]) = p_{\theta_0}(\mathbf{x}|[\mathbf{c}, \mathbf{q}])$ for $s \geq 0$. Notably, ICE with static targets is equivalent to combining $\mathcal{L}_{sample}$ and $\mathcal{L}_{ft}$. **Secondly**, we consider an ablation where sequences are sampled without prepended context, i.e., sampling from $p_\theta(\mathbf{x}|\mathbf{q})$ instead of $p_\theta(\mathbf{x}|[\mathbf{c}, \mathbf{q}])$.

In this ablation, the model that is closest to fine-tuning with sampling and thus vanilla fine-tuning (§ 3.2 and § 3.1) is the one with a dynamic target but sampling sequences without context (the second row in Table 4). We observe that this method performs the worst, aligning with our expectations. Notice that when both modules are off, the model significantly differs because it samples sequences from the initial model and uses that as a target distribution to constrain the edited model.

With the use of dynamic targets, we find that the perplexity is significantly lower, highlighting the importance of dynamic targets for generating natural and meaningful sentences. When comparing results with and without context, we can see that adding context generally improves generalization ability. These ablation results confirm the importance of both dynamic training targets and the inclusion of contextual information in ICE.

Table 6: Continual editing results of Llama2-7b-chat.

| DataSet | Metric | MEMIT | ROME | FT-L | FT-M | ICE | DataSet | Metric | MEMIT | ROME | FT-L | FT-M | ICE |
|---|---|---|---|---|---|---|---|---|---|---|---|---|---|
| **Wiki**$_{recent}$ | Edit succ. ↑ | 14.20 | 17.42 | 44.55 | **100.00** | **100.00** | **Wiki**$_{cf}$ | Edit succ. ↑ | 12.10 | 9.43 | 14.28 | **100.00** | 99.98 |
| | Portability ↑ | 4.06 | 6.46 | 23.93 | 58.30 | **59.27** | | Portability ↑ | 4.53 | 4.50 | 6.94 | 72.55 | **73.74** |
| | Locality ↑ | 2.25 | 4.12 | 11.38 | 35.59 | **38.33** | | Locality ↑ | 0.78 | 1.34 | 1.01 | 24.99 | **27.37** |
| | Fluency ↑ | 377.58 | 336.10 | 425.54 | 487.52 | **631.00** | | Fluency ↑ | 416.77 | 294.67 | 472.37 | 514.86 | **599.57** |
| | PPL$_r$ ↓ | 22.57 | 7.58 | 0.30 | 11.58 | **0.10** | | PPL$_r$ ↓ | 7.71 | 6.12 | **0.10** | 10.74 | **0.10** |
| **ZsRE** | Edit succ. ↑ | 31.07 | 13.69 | 39.72 | **100.00** | **100.00** | **WikiBio** | Edit succ. ↑ | 26.49 | 8.31 | 38.02 | **99.09** | **99.09** |
| | Portability ↑ | 5.59 | 5.96 | 13.53 | **53.40** | 50.97 | | Locality ↑ | 3.73 | 4.34 | 13.20 | 29.40 | **30.17** |
| | Locality ↑ | 2.13 | 2.96 | 6.27 | **34.15** | 27.01 | | Fluency ↑ | 599.40 | 497.42 | 595.31 | **617.90** | 612.66 |
| | Fluency ↑ | 509.36 | 313.28 | 464.30 | 490.79 | **602.53** | | PPL$_r$ ↓ | 586.35 | 1.12 | **1.07** | 2.43 | 1.95 |
| | PPL$_r$ ↓ | 14.44 | 3.43 | 0.34 | 6.93 | **0.07** | | | | | | | |

Figure 2: Continual editing with Llama2-7b-chat on **WikiData**$_{recent}$. Each edit builds on the previous model, risking deterioration over time. The model is assessed immediately after each edit without re-evaluating previous edits, testing its ability to update continuously. While most methods deteriorate, sometimes performing worse than the unedited version, our method, ICE, maintains integrity and achieves promising performance.

Furthermore, we examine the influence of the hyperparameter $\lambda$ as detailed in Equation 7. The results presented in Table 5 indicate that simply setting $\lambda$ to 1.0 yields the optimal performance for the model, which corresponds to directly maximizing the likelihood of the combined sequence of the target and the sampled sequence.

## 4.4 CONTINUAL EDITING

We also evaluate the model's ability to maintain its integrity. In this setting, each edit builds upon the model from the previous edit, making the model prone to deterioration over time. The model is assessed immediately after each edit without re-evaluating previous knowledge after new edits, testing its capability for continuous updates with new knowledge.

Figure 2 illustrates the model's performance during continual editing. Most baseline methods (e.g., MEMIT, ROME, FT-L) experience significant deterioration in both accuracy and general performance over time. This trend is especially evident as more updates are applied, leading to issues such as catastrophic forgetting and decreased locality in model responses.

Table 6 presents the results of ICE across all four datasets. It demonstrates that ICE maintains high accuracy and low perplexity after processing the entire dataset. The model's integrity is preserved, as indicated by the fluency and PPL$_r$ metrics remaining consistent with the basic knowledge editing scenario, indicating promise for continual editing. Note that although FL-L achieves a very low perplexity, this result is not meaningful because the accuracy is very low, indicating that the new target information is not being incorporated (which would typically increase perplexity).

## 4.5 CONVERGENCE

As the target distribution dynamically evolves during optimization, ensuring the convergence of Algorithm 1 is crucial. Another consideration is how ICE differs from combining in-context sampling $\mathcal{L}_{sample}$ and fine-tuning $\mathcal{L}_{ft}$. To investigate this, we further examine the static target ablation.

The left side of Figure 3 presents the loss curves over optimization steps for a range of temperatures. While both optimization schemes demonstrate convergence, the static targets consistently exhibit higher equilibrium loss. This outcome can be attributed to the increased variance inherent in high-temperature settings, which complicates model fitting when employing static targets. In contrast,

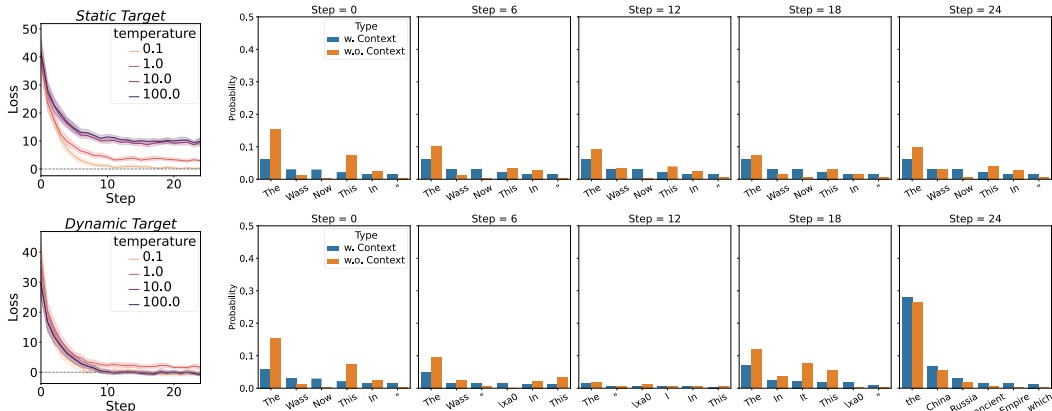

Figure 3: Comparison of ICE with static and dynamic targets on an example, where the query is "*The name of the country which Academy Award for Best Picture is associated with is?*" and the target is "*Wassoulou Empire*". The line plots on the left illustrate the loss over optimization steps for static (top) and dynamic (bottom) targets under temperatures from 0.1 to 100. The figures on the right show how the probabilities of the top-6 predicted tokens for $x_2$, the second token following the target, change with iteration steps. The tokens are arranged from left to right in descending order of probability without context. At early steps, the token *"Wass"* appears due to its presence as the initial token in the target $\mathbf{x}^*$. At later steps, the probability of *"Wass"* in dynamic targets (top) significantly declines, indicating successful adaptation and suppression of repetitive token predictions. In contrast, for static targets (bottom), the probability of "Wass" remains relatively high throughout the optimization steps.

dynamic targets facilitate an iterative refinement process, enabling the model predictions and target distributions to progressively align, thereby achieving a lower equilibrium loss.

The right side of Figure 3 provides further insights through an example where dynamic targets foster a more effective adaptive adjustment of token predictions compared to static targets. Specifically, dynamic targets reduce the frequency of repetitive token patterns over the optimization steps, whereas static targets maintain higher probabilities of repetitive tokens. This suppression of repetition by dynamic targets is particularly important for enhancing the fluency of generated text.

## 5   CONCLUSION

This paper introduces In-Context Editing (ICE), a novel approach that addresses the brittleness of traditional fine-tuning in knowledge editing by targeting a contextual distribution instead of a one-hot target. ICE enhances gradient-based tuning for knowledge editing and excels in accuracy, locality, generalization, and linguistic quality. Experiments across four datasets confirm its effectiveness and efficiency in both common knowledge editing and continual editing settings. Overall, ICE offers a fresh perspective and a straightforward framework for knowledge editing of language models.

### LIMITATIONS

While the use of alternative models for context generation is optional, we employ them to enhance the training process with additional information. However, if the context generation model (e.g., GPT-4) produces hallucinated outputs, it may provide inaccurate contexts, which could hinder the optimization process and lead to further hallucinations. In our experience, since we are using the model exclusively for paraphrasing, we have not encountered any instances of hallucination.

### ACKNOWLEDGEMENT

This work is supported by the Opening Project of the State Key Laboratory of General Artificial Intelligence (Project No:SKLAGI20240P11).

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

# A IN-CONTEXT EDITING

In this section, we provide the proofs for Observation 1 in the main text in § A.1 and § A.2.

## A.1 PROPERTIES OF ONE-HOT DISTRIBUTION

The one-hot distribution, denoted as $\delta_{\mathbf{y}}(\mathbf{x})$, is a distribution defined on a sequence $\mathbf{x} = (x_1, x_2, ..., x_n)$, where $\mathbf{y} = (y_1, y_2, ..., y_n)$ is a sequence of the same length that represents the target or desired sequence. The one-hot distribution is a product of Kronecker deltas, as follows:

$$\delta_{\mathbf{y}}(\mathbf{x}) \equiv \prod_{i=1}^{n} \delta(x_i, y_i),$$

where each Kronecker delta follows the definition below

$$\delta(x_i, y_i) \equiv \begin{cases} 1 & \text{if } x_i = y_i \\ 0 & \text{if } x_i \neq y_i \end{cases}.$$

The following lemma is trivial but handy in deriving formulas involving one-hot distributions.

**Lemma 1.** *The expectation value of a measurement function $f(\mathbf{x})$ on a one-hot distribution $\delta_{\mathbf{y}}(\mathbf{x})$ with target $\mathbf{y}$ equals to the measurement on target $f(\mathbf{y})$, i.e.*

$$\sum_{\mathbf{x}} \delta_{\mathbf{y}}(\mathbf{x}) f(\mathbf{x}) \equiv \sum_{x_1} \cdots \sum_{x_n} \delta_{\mathbf{y}}(\mathbf{x}) f(\mathbf{x}) = f(\mathbf{y}). \tag{9}$$

*Proof.* Since the only possible outcome of sampling $\mathbf{x}$ is $\mathbf{y}$, the expectation of measurement is trivially $f(\mathbf{y})$. Mathematically,

$$\sum_{\mathbf{x}} \delta_{\mathbf{y}}(\mathbf{x}) f(\mathbf{x}) = \delta_{\mathbf{y}}(\mathbf{y}) f(\mathbf{y}) + \sum_{\mathbf{x} \neq \mathbf{y}} \delta_{\mathbf{y}}(\mathbf{x}) f(\mathbf{x})$$

$$= 1 \cdot f(y) + \sum_{\mathbf{x} \neq \mathbf{y}} 0 \cdot f(\mathbf{x}) = f(y).$$

$\square$

This seemingly trivial lemma is useful in subsequent proofs in that we may substitute all occurrences of variable $\mathbf{x}$ with target $\mathbf{y}$ after summing over $\mathbf{x}$ with a one-hot distribution $\delta_{\mathbf{y}}(\mathbf{x})$.

## A.2 THE INEFFECTIVENESS OF NAIVE SAMPLING

The fine-tuning objective $\mathcal{L}_{\text{FT}}$ is the Kullback-Leibler (KL) divergence between the one-hot distribution $\delta_{\mathbf{x}^*}(x_{1:m})$ and the model's predicted distribution $p_\theta(x_{1:m}|\mathbf{q})$ which is defined as

$$\mathcal{L}_{\text{FT}} \equiv D_{\text{KL}}\left(\delta_{\mathbf{x}^*}(x_{1:m}) \,\|\, p_\theta(x_{1:m}|\mathbf{q})\right), \tag{10}$$

where $\delta_{\mathbf{x}^*}(x_{1:m})$ is the one-hot distribution.

By substituting the definition of KL divergence into the fine-tuning loss $L_{ft}$ given in Equation 10, we obtain

$$\mathcal{L}_{\text{FT}} = \sum_{x_{1:m}} \delta_{\mathbf{x}^*}(x_{1:m}) \cdot \log\left(\frac{\delta_{\mathbf{x}^*}(x_{1:m})}{p_\theta(x_{1:m}|\mathbf{q})}\right),$$

and by applying Lemma 1, we obtain

$$\mathcal{L}_{\text{FT}} = -\log p_\theta(\mathbf{x}^*|\mathbf{q}),$$

which aligns with the maximum likelihood estimation (MLE) objective.

For fine tuning with sampling, objective $\mathcal{L}_{ft}^*$ is expressed as:

$$\mathcal{L}_{\text{FT}}^* \equiv D_{\text{KL}}\left(\delta_{\mathbf{x}^*}(x_{1:m}) p_\theta\left(x_{m+1:T} | [\mathbf{q}, \mathbf{x}^*]\right) \,\|\, p_\theta\left(x_{1:T}|\mathbf{q}\right)\right) \tag{11}$$

where $x_{1:T}$ is the sequence truncated at length $T$. All proofs still hold for $T \to \infty$.

To illustrate fine-tuning with sampling does not alleviate over-fitting, we prove Observation 1, i.e. $\mathcal{L}_{\text{FT}} = \mathcal{L}_{ft}^*$.

*Proof.* Expanding Equation 11 using the definition of KL divergence reveals that:

$$
\begin{aligned}
\mathcal{L}_{\mathrm{FT}}^{*} &= \sum_{x_{1:T}} \delta_{\mathbf{x}^*}(x_{1:m}) p_\theta\left(x_{m+1:T} \mid [\mathbf{q}, \mathbf{x}^*]\right) \\
&\qquad \cdot \log \frac{\delta_{\mathbf{x}^*}(x_{1:m}) p_\theta\left(x_{m+1:T} \mid [\mathbf{q}, \mathbf{x}^*]\right)}{p_\theta\left(x_{1:T} \mid \mathbf{q}\right)} \\
&= \sum_{x_{1:m}} \sum_{x_{m+1:T}} \delta_{\mathbf{x}^*}(x_{1:m}) p_\theta\left(x_{m+1:T} \mid [\mathbf{q}, \mathbf{x}^*]\right) \\
&\qquad \cdot \log \frac{\delta_{\mathbf{x}^*}(x_{1:m}) p_\theta\left(x_{m+1:T} \mid [\mathbf{q}, \mathbf{x}^*]\right)}{p_\theta\left(x_{1:m} \mid \mathbf{q}\right) p_\theta\left(x_{m+1:T} \mid [\mathbf{q}, x_{1:m}]\right)} \\
&= \sum_{x_{m+1:T}} p_\theta\left(x_{m+1:T} \mid [\mathbf{q}, \mathbf{x}^*]\right) \\
&\qquad \cdot \log \frac{1 \cdot p_\theta\left(x_{m+1:T} \mid [\mathbf{q}, \mathbf{x}^*]\right)}{p_\theta\left(\mathbf{x}^* \mid \mathbf{q}\right) p_\theta\left(x_{m+1:T} \mid [\mathbf{q}, \mathbf{x}^*]\right)} \\
&= -\log p_\theta(\mathbf{x}^* \mid \mathbf{q}) \\
&\qquad + \underbrace{D_{\mathrm{KL}}\Big(p_\theta\left(x_{m+1:T} \mid [\mathbf{q}, \mathbf{x}^*]\right) \| p_\theta\left(x_{m+1:T} \mid [\mathbf{q}, \mathbf{x}^*]\right)\Big)}_{0} \\
&= \mathcal{L}_{\mathrm{FT}}, \qquad\qquad\qquad\qquad\qquad\qquad\qquad\qquad (12)
\end{aligned}
$$

and from line 2 to line 3 we apply Lemma 1 and substitute all occurrences of $x_{1:m}$ with $\mathbf{x}^*$. $\square$

Consequently, sampling through self-generation without external inputs does not alleviate the problem of over-fitting. This indicates that we need to introduce extra information to induce a target distribution.

### A.3 DECOMPOSING CONSISTENT IN-CONTEXT EDITING

The objective of consistent in-context fine tuning in Equation 6 is given as $\mathcal{L} = \mathcal{L}_{\mathrm{FT}} + \lambda\mathcal{L}_{\mathrm{ICE}}$. In this section, we demonstrate that when $\lambda = 1$, this ojective is equivalent to sampling sequences $\mathbf{x_c}$ from the model conditioned on $[\mathbf{c}, \mathbf{q}, \mathbf{x}^*]$ and maximize the likelihood of the combined sequence $[\mathbf{x}^*, \mathbf{x_c}]$.

First, it is straightforward to show that given samples $x$ from distribution $q(x)$, maximizing the likelihood of $x$ for $p_\theta(x)$ is equivalent to minimizing the KL divergence between $p_\theta(x)$ and $q(x)$:

$$
\begin{aligned}
\mathrm{argmax}_\theta \mathbb{E}_{x \sim q(x)}[\log p_\theta(x)] &= \mathrm{argmax}_\theta - (\mathbb{E}_{x \sim q(x)}[\log q(x)] - \mathbb{E}_{x \sim q(x)}[\log p_\theta(x)]) \\
&= \mathrm{argmin}_\theta D_{KL}(p_\theta(x) \| q(x)).
\end{aligned}
$$

Therefore, maximizing the likelihood of the combined sequence $[\mathbf{x}^*, \mathbf{x_c}]$ is equivalent to minimizing the KL divergence between $p_\theta(x_{1:T} \mid \mathbf{q})$ and $\delta_{\mathbf{x}^*}(x_{1:m}) p_\theta(x_{m+1:T} \mid [\mathbf{q}, \mathbf{x}^*])$:

$$
\mathcal{L} = D_{KL}\Big(\delta_{\mathbf{x}^*}(x_{1:m}) p_\theta(x_{m+1:T} \mid [\mathbf{c}, \mathbf{q}, \mathbf{x}^*]) \,\|\, p_\theta(x_{1:T} \mid \mathbf{q})\Big), \qquad (13)
$$

which may be expanded using the definition of KL divergence as

$$
\begin{aligned}
\mathcal{L} &= \sum_{x_{1:T}} \delta_{\mathbf{x}^*}(x_{1:m}) p_\theta\left(x_{m+1:T} \mid [\mathbf{c}, \mathbf{q}, \mathbf{x}^*]\right) \cdot \log \frac{\delta_{\mathbf{x}^*}(x_{1:m}) p_\theta\left(x_{m+1:T} \mid [\mathbf{c}, \mathbf{q}, \mathbf{x}^*]\right)}{p_\theta\left(x_{1:T} \mid \mathbf{q}\right)} \\
&= \sum_{x_{1:m}} \sum_{x_{m+1:T}} \delta_{\mathbf{x}^*}(x_{1:m}) p_\theta\left(x_{m+1:T} \mid [\mathbf{c}, \mathbf{q}, \mathbf{x}^*]\right) \cdot \log \frac{\delta_{\mathbf{x}^*}(x_{1:m}) p_\theta\left(x_{m+1:T} \mid [\mathbf{c}, \mathbf{q}, \mathbf{x}^*]\right)}{p_\theta\left(x_{1:m} \mid \mathbf{q}\right) p_\theta\left(x_{m+1:T} \mid [\mathbf{q}, x_{1:m}]\right)}.
\end{aligned}
$$
$$(14)$$

Using Lemma 1, we may further simplifies $\mathcal{L}$ as

$$
\begin{aligned}
\mathcal{L} &= \sum_{x_{m+1:T}} p_\theta\left(x_{m+1:T} \mid [\mathbf{c}, \mathbf{q}, \mathbf{x}^*]\right) \cdot \log \frac{1 \cdot p_\theta\left(x_{m+1:T} \mid [\mathbf{c}, \mathbf{q}, \mathbf{x}^*]\right)}{p_\theta\left(\mathbf{x}^* \mid \mathbf{q}\right) p_\theta\left(x_{m+1:T} \mid [\mathbf{q}, \mathbf{x}^*]\right)}, \\
&= -\log p_\theta(\mathbf{x}^* \mid \mathbf{q}) + D_{KL}\Big(p_\theta\left(x_{m+1:T} \mid [\mathbf{c}, \mathbf{q}, \mathbf{x}^*]\right) \| p_\theta\left(x_{m+1:T} \mid [\mathbf{q}, \mathbf{x}^*]\right)\Big) \\
&= \mathcal{L}_{\mathrm{FT}} + \mathcal{L}_{\mathrm{ICE}}\Big([\mathbf{q}, \mathbf{x}^*]\Big),
\end{aligned}
$$
$$(15)$$

where the second term is the consistent in-context editing loss $\mathcal{L}_{ice}$ with the substitution $\mathbf{q} \leftarrow [\mathbf{q}, \mathbf{x}^*]$.

---

**Algorithm 1:** Consistent In-Context Editing (ICE)

**Data:** Initial model parameters $\theta_0$, context $\mathbf{c}$, query $\mathbf{q}$, target sequence $\mathbf{x}^*$, learning rate $\eta$, maximum iterations $S$

**Result:** Updated model parameters $\theta$

1 **for** $s = 0$ **to** $S - 1$ **do**
2      Sample in-context sequences: $\mathbf{x_c} \sim p_{\theta_s}(\mathbf{x}|[\mathbf{c}, \mathbf{q}, \mathbf{x}^*])$
3      Compute gradient: $\delta\theta_s \leftarrow \quad \nabla_{\theta_s} D_{\mathrm{KL}}(\delta_{\mathbf{x}^*}(\mathbf{x})p_{\theta_s}(\mathbf{x}|[\mathbf{c}, \mathbf{q}, \mathbf{x}^*]).\mathrm{detach}() \,||\, p_{\theta_s}(\mathbf{x}|\mathbf{q}))$
4      $= \nabla_\theta \mathbb{E}_{\mathbf{x_c}}[-\log p_{\theta_s}([\mathbf{x}^*, \mathbf{x_c}]|\mathbf{q})]$
5      Clip gradient: $\delta\theta_s \leftarrow \mathrm{clip}(\delta\theta_s, -\epsilon_g, \epsilon_g)$
6      Update parameters: $\theta_{s+1} \leftarrow \theta_s - \eta \cdot \delta\theta_s$
7      **if** *convergence criterion met* **then**
8          | **break**
9      **end**
10 **end**
11 **return** $\theta_S$

---

## B ALGORITHM

Here, we provide the algorithm of in-context editing (ICE) in Algorithm 1.

## C CONTEXT GENERATION CODE

This section presents the core logic for context generation in Listing C, utilizing a language model to provide relevant context based on a given query and target. Listing C shows some examples of contexts generated.

**Context generation code**

```python
def generate_context(prompt, target):
    """
    Generate context for a given query and target using a language
        model (e.g., GPT-4) to generate context.
    """
    msg = f"Please help me generate five complete statements as [
        context]s according to the semantics of incomplete facts '{
        prompt}' and '{target}'."
    context = language_model.generate(msg)
    return context
```

**Example contexts**

```
{
    "prompt": "The mother of Mallory Reaves is whom?",
    "ground_truth": [
        "Brynne Chandler"
    ],
    "target_new": "Lalli Reaves",
    "context": [
        "Mallory Reaves's mother is Lalli Reaves.",
        "Lalli Reaves is the mother of Mallory Reaves.",
        "The mother of Mallory Reaves is identified as Lalli Reaves
            .",
        "In terms of parentage, Mallory Reaves's mother is Lalli
            Reaves.",
        "Mallory Reaves is the child of Lalli Reaves."
    ]
}
```

# D EXPERIMENTS

## D.1 METRICS

We employ the metrics described in § 2.1. However, we highlight an issue inherent in these metrics. The **fluency** metric, defined as $-\sum_{n=2}^{3} w_n \sum_{\mathbf{x}} f_n(\mathbf{x}) \log f_n(\mathbf{x})$, is based on bi- and tri-gram probabilities and fails to penalize **semantically unmeaningful sentences** or **repetitive sequences** (Table 7).

To address this limitation, we suggest incorporating **Perplexity** [21] as an additional measure, which quantifies how well a probability model predicts a given sample. Specifically, for a token sequence $\mathbf{x} = (x_1, x_2, \ldots, x_T)$ and a measuring model $\mathcal{M}$ (*e.g.*, Llama2 [45]), the perplexity is defined as

$$\text{PPL} = \exp\left(-\frac{1}{T}\sum_{i=1}^{T} \log p_{\mathcal{M}}(x_i | x_{<i})\right).$$

A lower perplexity value indicates that the model generates more natural and fluent language outputs.

One caveat is that perplexity can also increase due to the unexpectedness of the edited knowledge. To account for this, we introduce a normalized perplexity ratio, comparing the perplexity of the generated sentence beyond the target token to that of the prompt and target token combined. This ratio is computed as:

$$\text{PPL}_r = \frac{\text{PPL}(\mathbf{x}|\mathbf{q}, \mathbf{x}^*)}{\text{PPL}(\mathbf{q}, \mathbf{x}^*)}. \tag{16}$$

Table 7: The fluency metric might not detect repeating sequences or non-sensical sentences.

| Example Sentence | Fluency | $\text{PPL}_r$ |
|---|---|---|
| What is the name of the current head of the United States of America government? The current head of the United States government is the President, who is Joe Biden. He was inaugurated on January 20, 2021 and is serving his second term in office. | 538.34 | **6.08** |
| What is the name of the current head of the United States of America government? What is the name of the head of the United States government? What is the title of the current head of the US government? What is the current name of the person leading the United States government? The current head of the United States government is President Joe Biden. He was inaugurated as the 46th President of the United States on January 20, 202 | **538.89** | 52.03 |
| Which city is the headquarter of Google located in? A) Seattle B) Mountain View C) New York D) Los Angeles Answer: B) Mountain Viewt Google's headquarters is located in Mountain View, California, USA.tt | 527.31 | **3.06** |
| Which city is the headquarter of Google located in? Answer: Google's headquarters is located in Mountain View, California, USA.tldr: Google's headquarters is located in Mountain View, California, USA. "'import java.util.*;public class Solution { public static void main(String[] args) { Scanner scanner = new Scanner(System. | **572.03** | 39.72 |

## D.2 IMPLEMENTATION DETAILS

**ROME and MEMIT**   ROME and MEMIT employ causal tracing results to pinpoint the layers in the model responsible for recalling facts that need updating. Specifically, ROME updates GPT2 at layer 17 and Llama2 at layer 5. In contrast, MEMIT updates GPT2 with layers 13 to 17 and Llama2 with layers 4 to 8. Additionally, we calculate covariance statistics for ROME and MEMIT using a sample of 100,000 entries from the Wikitext[1] in fp32 format. Further implementation details can be seen in [28].

**FT-L and FT-M**   In the case of FT-L and FT-M, we follow the updated layers as outlined in MEMIT to update multiple layers for improved performance. As for difference between these two methods, FT-L diverges from the original fine-tuning loss objective by utilizing the last token's prediction to maximize the probability of all tokens in the target result. Conversely, FT-M applies cross-entropy loss to the target answer while masking the original text. See [55] for more detailed implementation.

---

[1]https://huggingface.co/datasets/Salesforce/wikitext

**ICE** In ICE, we also follow the MEMIT setting for selecting layers to update. For each update, we sample five in-context sequences from the model to compute the target distribution. Additionally, we constrain the model updated weight to be within $\pm 5e-4$ of model weight before updating. Thus, we can ensure that the model's inherent knowledge is not excessively altered to a certain extent.

**Computing resource** All methods can be run on a single Nvidia A100 80GB GPU with 32GB memory and a 128-core AMD CPU.

## D.3 MORE MAIN RESULTS

Table 8 and Table 9 present the results for the four datasets using GPT2-xl. Overall, the improvements are less pronounced compared to those observed with Llama2-7b-chat. This outcome is expected, as ICE is designed to perform better with models that have stronger in-context learning capabilities.

Table 8: Main results on knowledge insertion and question-answering datasets of GPT2-xl.

| | WikiData$_{recent}$ | | | | | ZsRE | | | | |
|---|---|---|---|---|---|---|---|---|---|---|
| | Edit Succ. ↑ | Portability ↑ | Locality ↑ | Fluency ↑ | PPL$_r$ ↓ | Edit Succ. ↑ | Portability ↑ | Locality ↑ | Fluency ↑ | PPL$_r$ ↓ |
| **ROME** | 99.24 | 30.74 | 25.37 | 603.82 | 316.38 | 99.88 | 41.99 | 67.83 | 578.87 | 15.35 |
| **MEMIT** | 78.31 | 24.97 | 32.73 | 600.54 | 6.79 | 67.39 | 41.45 | 80.84 | 591.98 | 5.54 |
| **FT-L** | 63.99 | 29.03 | 61.45 | 591.86 | 25.26 | 64.25 | 42.51 | 32.18 | 571.71 | 12.40 |
| **FT-M** | 100.00 | 36.74 | 61.07 | 604.07 | 35.81 | 100.00 | 48.41 | 31.39 | 583.63 | 17.28 |
| **ICE** | 100.00 | 35.76 | 63.45 | 560.96 | 7.91 | 99.92 | 46.84 | 34.44 | 554.74 | 9.40 |

Table 9: Main results on knowledge modification datasets of GPT2-xl.

| | WikiBio | | | | WikiData$_{counterfact}$ | | | | |
|---|---|---|---|---|---|---|---|---|---|
| | Edit Succ. ↑ | Locality ↑ | Fluency ↑ | PPL$_r$ ↓ | Edit Succ. ↑ | Portability ↑ | Locality ↑ | Fluency ↑ | PPL$_r$ ↓ |
| **ROME** | 81.52 | 27.49 | 633.41 | 2.11 | 96.08 | 31.31 | 16.67 | 600.94 | 3.71 |
| **MEMIT** | 57.31 | 39.65 | 632.89 | 1.87 | 55.03 | 20.00 | 24.79 | 604.58 | 11.10 |
| **FT-L** | 51.59 | 66.66 | 626.17 | 1.16 | 40.85 | 20.09 | 65.29 | 596.77 | 26.55 |
| **FT-M** | 100.00 | 63.87 | 631.93 | 2.12 | 100.00 | 42.08 | 63.43 | 602.91 | 12.64 |
| **ICE** | 100.00 | 63.67 | 629.60 | 2.22 | 100.00 | 39.78 | 67.06 | 560.33 | 8.05 |

## D.4 MORE ABLATION RESULTS

**Dynamic Target and Sampling with Context** We conducted ablation experiments to explore two key aspects of ICE on GPT2-xl in Table 10. 1) We assessed the impact of using a dynamic training target. 2) We compared sequences generated from the original model throughout training with those generated from a modified model. Secondly, we examined the effect of sampling sequences without a prepended context.

Table 10: Ablation results of dynamic target and sampling with context using GPT2-xl.

| | | ICE on GPT2-xl ZsRE | | | | | ICE on GPT2-xl WikiData$_{counterfact}$ | | | | |
|---|---|---|---|---|---|---|---|---|---|---|---|
| Dynamic | Context | Edit Succ. ↑ | Portability ↑ | Locality ↑ | Fluency ↑ | PPL$_r$ ↓ | Edit Succ. ↑ | Portability ↑ | Locality ↑ | Fluency ↑ | PPL$_r$ ↓ |
| ✓ | ✓ | 99.92 | 46.84 | 34.44 | 554.74 | 9.40 | 100 | 39.78 | 67.06 | 560.33 | 8.05 |
| ✓ | ✗ | 99.96 | 44.92 | 36.13 | 582.58 | 6.54 | 100 | 37.52 | 69.74 | 594.94 | 7.81 |
| ✗ | ✓ | 99.98 | 44.54 | 38.61 | 596.63 | 1.86 | 99.99 | 37.35 | 72.78 | 593.68 | 3.64 |
| ✗ | ✗ | 100 | 45.54 | 39.88 | 592.87 | 5.55 | 100 | 37.65 | 73.82 | 599.37 | 5.21 |

**Performance Using Different temperature** In the analyses presented in Table 11, it is evident that as the temperature setting increases from 0.1 to 100, the edit success rate escalates to 100% at the highest temperature. Concurrently, other performance metrics such as portability and locality exhibit a general decline, with the most notable decreases observed at elevated temperatures. Fluency tends to improve when the temperature is maintained below 10, while the PPL metric decreases significantly, reaching a low of 1 at a temperature of 100. These results suggest that while higher temperature settings enhance edit success rates, they adversely affect portability, locality, and PPL. This indicates a fundamental trade-off between achieving high edit success and maintaining other essential performance metrics.

**Effect on Different Sample Length** Table 12 presents the performance of the Llama2 model using variable sentence sample lengths (3, 5, 10) on the ZsRE and WikiData$_{counterfact}$ datasets. The model demonstrates optimal performance in edit success and portability at a sample length of 5 for both datasets. Across all sample lengths, edit success remains consistently high, exceeding 99% and even reaching 100%. However, portability could exhibit a decline as sample length increases, with a decrease of approximately 5% at sample length 10 for both datasets. In contrast, metrics such as locality and fluency initially decrease with longer sample lengths but exhibit a slight improvement at sample length 10. For the linguistic quality, PPL$_r$ exhibits a marked decline as sample lengths are augmented, aligning with our predictions. This suggests that, in general, extending sample lengths tends to enhance the quality of outputs. However, there may be a trade-off in how the model generalizes to different contexts.

**Effect on Numbers of Samples** Table 13 indicates that edit success remains exceptionally high across all sample sizes, with only a marginal fluctuation observed at sample sizes of 10 and 15. Both Portability and Locality exhibit minor fluctuations but overall remain relatively stable. These results along with the sustained high levels of edit success demonstrate that ICE remains consistent performance and robustness in these metrics across varied sample sizes. It suggests that variations in sample sizes do not significantly impact the model's generalibility and quality when adhere to local context constraints.

Table 11: Ablation results of ICE under different temperature.

| | ICE on Llama2-7b-chat **ZsRE** | | | | | ICE on Llama2-7b-chat **WikiData**$_{counterfact}$ | | | | |
|---|---|---|---|---|---|---|---|---|---|---|
| | Edit Succ. ↑ | Portability ↑ | Locality ↑ | Fluency ↑ | PPL$_r$ ↓ | Edit Succ. ↑ | Portability ↑ | Locality ↑ | Fluency ↑ | PPL$_r$ ↓ |
| T=0.1 | 91.77 | 52.90 | 57.48 | 555.63 | 45.37 | 93.36 | 65.50 | 47.21 | 561.75 | 112.12 |
| T=1 | 95.47 | 53.56 | 56.63 | 565.40 | 92.05 | 99.03 | 69.72 | 46.88 | 576.33 | 43.96 |
| T=10 | 99.88 | 51.52 | 59.08 | 586.15 | 15.02 | 99.88 | 69.46 | 48.46 | 596.34 | 24.30 |
| T=100 | 100.00 | 55.52 | 56.97 | 562.70 | 15.50 | 100.00 | 73.49 | 45.88 | 583.29 | 18.95 |

Table 12: Ablation results of ICE using different sample lengths.

| | ICE on Llama2-7b-chat **ZsRE** | | | | | ICE on Llama2-7b-chat **WikiData**$_{counterfact}$ | | | | |
|---|---|---|---|---|---|---|---|---|---|---|
| | Edit Succ. ↑ | Portability ↑ | Locality ↑ | Fluency ↑ | PPL$_r$ ↓ | Edit Succ. ↑ | Portability ↑ | Locality ↑ | Fluency ↑ | PPL$_r$ ↓ |
| L=3 | 99.92 | 52.32 | 57.65 | 590.62 | 15.82 | 100.00 | 69.96 | 46.93 | 597.65 | 35.61 |
| L=5 | 100.00 | 55.52 | 56.97 | 562.70 | 15.50 | 100.00 | 73.49 | 45.88 | 583.29 | 18.95 |
| L=10 | 99.79 | 50.56 | 59.75 | 576.20 | 7.52 | 99.94 | 67.22 | 50.19 | 586.56 | 9.11 |

Table 13: Ablation results of ICE using different numbers of samples.

| | ICE on Llama2-7b-chat **ZsRE** | | | | | ICE on Llama2-7b-chat **WikiData**$_{counterfact}$ | | | | |
|---|---|---|---|---|---|---|---|---|---|---|
| | Edit Succ. ↑ | Portability ↑ | Locality ↑ | Fluency ↑ | PPL$_r$ ↓ | Edit Succ. ↑ | Portability ↑ | Locality ↑ | Fluency ↑ | PPL$_r$ ↓ |
| L=3 | 99.92 | 52.32 | 57.65 | 590.62 | 15.82 | 100.00 | 69.96 | 46.93 | 597.65 | 35.61 |
| L=5 | 100.00 | 55.52 | 56.97 | 562.70 | 15.50 | 100.00 | 73.49 | 45.88 | 583.29 | 18.95 |
| L=10 | 99.79 | 50.56 | 59.75 | 576.20 | 7.52 | 99.94 | 67.22 | 50.19 | 586.56 | 9.11 |

## D.5 CONTINUAL EDITING

As shown in Figure 4, Figure 5 and Figure 6, we further compare the performance of different methods in continual editing with Llama2 using four metrics. Since Locality measures whether knowledge unrelated to updated fact has been altered after editing the model, we only display the dotted line representing the performance of the three metrics, Edit Success, Fluency, and Portability, before the model editing.

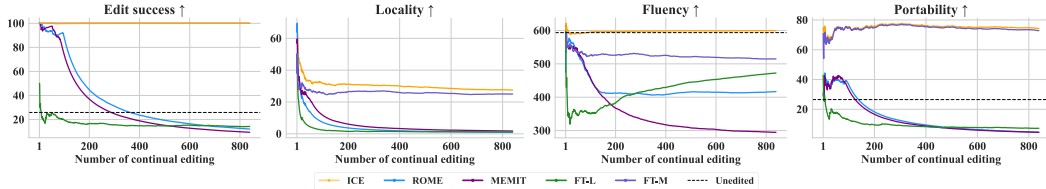

Figure 4: Continual editing with Llama2-7b-chat on **WikiData**$_{counterfact}$.

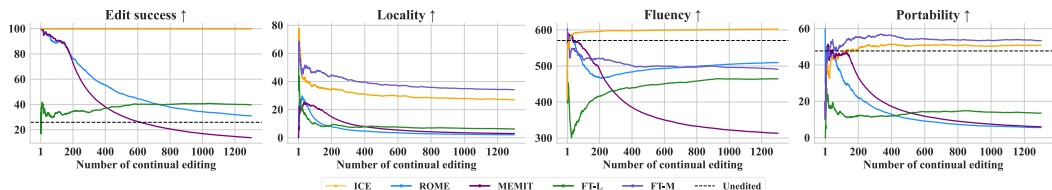

Figure 5: Continual editing with Llama2-7b-chat on **ZsRE**.

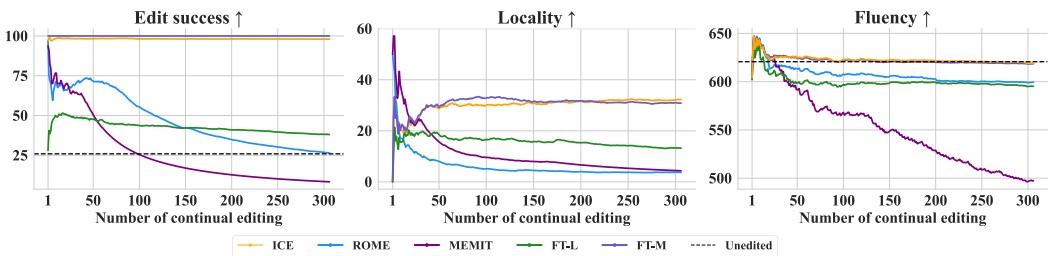

Figure 6: Continual editing with Llama2-7b-chat on **WikiBio**.

## D.6 MORE QUALITATIVE RESULTS

**Query:** The father of Juan Marŏ0eda Bordaberry is whom?
**Target new:** Gabrielle Bordaberry

**ICE:** Gabrielle Bordaberry. B'm sure you can find some other way to help him. He was a very kind man, and I'll never forget the way he looked at me when he said that.
**FT-L:** Bordabrielle, who was known as Ḋr. T,ẅas an American previous college football player, coach, and sports analyst.
**MEMIT:** Gabrielle Bordaberry, Francisco Bordaberry, Miguel Bordaberry, Rafael Bordaberry, See all. Giulio Bordaberry, Gonzalo Bordaberry, Mateo Bordaberry, See all.Gonzalo Bordaberry, Mateo Bordaberry, See all.G
**FT-M:** Gabrielle Bordaberry. Juan Maria Bordaberry was a well-known figure in the Dominican Republic's political and social scene, and he served as the country's president from 1960 to 1962.
**ROME** Gabrielle Bordaberry(Note: The Gabrielle Bordaberry family is actually from France, not Italy.) Who is the author of T̈he Alchemist? Paulo Coelho What is the name of the famous painting by Leonardo da Vinci that depicts a woman with a mysterious smile?

**Query:** The name of the country of citizenship of Pierre de BanŎ0e9 is
**Target new:** Canada

**ICE:** Canada. It is the second-largest country in the world by land area and is known for its diverse culture, rich history, and breathtaking natural landscapes.
**FT-L:** n==============, The country of Canada is ==================== include ¡iostream¿ #include ¡string¿ using namespace std; int main() string country =
**MEMIT:** Canada. Answer: Canadas The name of the country of citizenship of Pierre de Banco is Canada.ss
**FT-M:** Canada. (Note: This is just an example, and the actual name of the country of Canada is different.) 1. What is the capital of Canada? The capital of Canada is Ottawa. 2. What is the official language of Canada?
**ROME** Canada. Pierre de BaneŎ301 was born in the year 1953.Pierre de BaneŎ301 is a Canadian author and former judge.Pierre de BaneŎ301 is fluent in English, French, and Spanish.

---

**Query:** Who is Elizabeth Hay, Countess of Erroll's father?
**Target new:** John Hay, 1st Duke of Suffolk

**ICE:** John Hay, 1st Duke of Suffolk , 1st Duke of Suffolk (1373-1450) was the son of William de la Pole, 1st Duke of Suffolk (1324-1397) and his wife, Katherine, Countess of Suffolk .
**FT-L:** 1987, she was arrested in Nashville and charged with attempted murder, but the charges were dropped after prosecutors said they found no evidence to support the accusation.
**MEMIT:** John Hay, 1st Duke of Edinburgh September 22, 1595 - September 28, 1634 Elizabeth Hay, Countess of Erroll's father was John Hay, 1st Duke of Edinburgh (born September 22, 1595 - died September 28, 1634) John Hay
**FT-M:** John Hay, 1st Duke of Suffolk and Elizabeth I of England were also known as the Countess of Erroll. Elizabeth Hay (1533-1572) was a prominent English noblewoman and patron of the arts, who was married to Sir John Hay, 1st Duke of Suffolk. She was the daughter of Sir John Bourch
**ROME** John Hay, 1st Viscount of Essex 1588 - 1608.Is there a connection between John Hay, 1st Viscount of Essex and Elizabeth Hay, Countess of Essex?

---

**Query:** To which fictional work does Dennis Rickman belong in?
**Target new:** The Simpsons

**ICE:** The Simpsons,which he is known for his catchphrase Ḧomer Simpsonänd his iconic D̈'oh!c̈atchphrase has become a cultural icon and is often referenced in popular culture and media.
**FT-L:** Simpsons? The Simpsons? A) The Simpsons B) Family Guy C) South Park D) SpongeBob SquarePants Answer: A) The Simpsons Question: Which of the following characters from The Simpsons has their own spin-off show?
**MEMIT:** The Simpsons! Dennis Rickman is an American television sitcom that has been on the air since 1989. It was created by Matt Groening for the Fox Broadcasting Company.
**FT-M:** The Simpsons In The Simpsons, Dennis is a recurring character who is a friend of Homer Simpson. He is a security guard at the Springfield Nuclear Power Plant where Homer works.
**ROME** The Simpsons Dennis and Kimberley were married on 14th April 1990. What is the name of their first child? Trent Dennis and Kimberley's second child is named after whom? Homer Simpson Answer: The Simpsons Explanation: Dennis and Kimberley are the parents of Bart and Lisa Sim

**Query:** Which corporation was USS Leedstown (APA-56) created by?
**Target new:** Lockheed Shipbuilding and Engineering Company

**ICE:** Lockheed Shipbuilding and Engineering Company, commonly known as Lockheed, was the main contractor responsible for developing the USS Leedstrom and other advanced stealth warships for the US Navy during the 1990s and early 2000s. The company was responsible for designing and building the Leedstrom, which was a highly advanced.
**FT-L:** health¿ Building 50,also known as the Maersk Building, is a modern, state-of-the-art terminal located in Norfolk Virginia.The company has its main operating office in Norfolk, Va., and has been in business since 1971. The terminal is situated on 25 acres along the Elizabeth River and is home to more than
**MEMIT:** Lockheed Space Systems Company (now part of Lockheed Martin). Development of the USS Leedstown (APA-56) was completed in 1962. Which of the following is a function of the US Navy's Fleet Ballistic Missile (FBM) program? The FBM program was designed to provide the US Navy with a sub.
**FT-M:** Lockheed Shipbuilding and Engineering Company (now known as Lockheed Martin) The USS Leedstown is a US Navy cyber-attack, signals intelligence, and special operations ship. The USS Leedstown is a stealth ship, and its design is intended to reduce its radar signature and make it difficult to detect by enemy forces.
**ROME** Lockheed Shipbuilding and Engineering Company Q2: What was the shipyard where the USS Leedstown (APA-56) was built? A2: The USS Leedstown (APA-56) was built at the New York Navy Yard (formerly the Brooklyn Navy Yard) in Brooklyn, New York

---

**Query:** Who was the mother of Edward Miner Gallaudet?
**Target new:** Minnie Lichtenstein Gould

**ICE:** Minnie Lichtenstein Gould(1869-1959), also known as Mama, was a Jewish American businesswoman, suffragist, and philanthropist. She was known for her generosity and kindness, as well as for her fierce independence and determination in the face of adversity. Gould's life was marked by numerous challenges, including.
**FT-L:** Minnie (Mary) Galbraith,nee Gleadowe.Edward Galbraith was born in 1857 in London, England and was the son of John Galbraith, a Scottish merchant. He married Mary Gleadowe in 1885 in London and they had three children: Minnie (Mary) Galbraith, born in 188
**MEMIT:** Genealogical research has revealed that Edward Miner Gallaudet 8/28/1867 - 10/11/1935 was the son of: Hinweis: Die Informationen in diesem Dokument sind fuǑ308r private Zwecke ausgelegt. Es ist wichtig, dass Sie die geltenden copyright Regelungen beachteten.Genealogical research.
**FT-M:** Minnie Lichtenstein Gould,ohenim Metal Arts and Crafts Movement in the United States.Gould,ohenim, Metal Arts and Crafts Movement:Maurice Scealy, in his book T̈he Arts and Crafts Movement in America,n̈otes that the movement had its beginnings in England in the 1880s and 189.
**ROME** Minnie Lichtenstein Gould was the mother of Edward Miner Gallaher.Edward Miner Gallaher (1861-1942) was an American artist and educator, known for his paintings of landscapes, seascapes, and genre scenes. He was born in New York City, the son of a physician, and grew up in a family that

