# OpenReview forum: "In-Context Editing: Learning Knowledge from Self-Induced Distributions"
_ICLR.cc/2025/Conference — ICLR 2025 Poster_

### Official Review · Reviewer_c9Gw · 2024-11-01

**Soundness:** 3
**Presentation:** 3
**Contribution:** 3
**Rating:** 6
**Confidence:** 4

**Summary:**

The paper proposes a method of updating an LLM to incorporate some specific piece of new information.
The LLM prompt is prefixed with this new information and the output distribution is used as the target for training the same LLM without that additional context.
This loss is used as a regulariser, along with the main fine-tuning loss.
Results show good performance compared to the chosen baselines.

**Strengths:**

The idea is interesting and makes sense, as the probability distribution provides more detailed information compared to just a one-hot target.
Evaluation is performed on several different datasets and with a number of different metrics.

**Weaknesses:**

A crucial baseline is currently missing. There needs to be an evaluation of the exact same proposed model but with lambda set to 0. Using only the fine-tuning (FT) training loss.
This is important to understand what effect the proposed regularising objective has on the model. As far as I can see, this has not been reported in the paper at the moment.
FT-M and FT-L are reported but these differ from the vanilla FT objective and update only 1 specific layer in the model.
It is mentioned in the appendix that the proposed ICE model is trained by updating the same layers as MEMIT, which would be 5 layers.
There needs to be a baseline that updates the same layers as the proposed model using the same FT objective, with the only difference being that the proposed ICE loss component is turned off (lambda = 0).

The clarity of the paper could be improved.
For example, Section 3.2 seems to propose a method but the actual mechanics or motivation are unclear to me. And then it is said that actually this method is equivalent to the vanilla method described in the previous section anyway.

The novelty of the method is somewhat overstated in the paper. The technical solution is essentially the same as previous work such as Snell et al (2022), the main thing changed is the content of the prompt. That paper is indeed referenced but only among a list of different directions. The particular novel aspect of the paper should be made clear and previous work should be attributed accordingly. As far as I can see, the novel aspect is the application of this method to updating facts in the LLM.

---------------------------
Updated after discussion:

It seems the missing baseline is actually included, it was just not clear from the presentation. I have increased my score accordingly. Although it is still a bit unclear which of the two baselines is then the lambda=0 equivalent.

Regarding novelty compared to Snell et al (2022): Even though you frame this conceptually as a regulariser here, the practical method is still essentially the same. The author response did not present any specific differences. This previous work should be highlighted accordingly in the paper.

**Questions:**

What is the value of lambda used? Was a different value used for different datasets? How was this value found?

Why is the WikiBio dataset the only one missing the Portability score?

Perplexity is reported as an evaluation metric but is that the perplexity of the trained model itself or the perplexity of some other reference model on the generated text?

It is said that one of the baselines "demonstrated nearly the best performance" in a survey. Why was the best model not reported as a baseline?

The information about which layers are updated by the proposed method should be in the main paper, not hidden deep in the appendix. The main paper currently only mentions this information for "other baselines" and the proposed model does not qualify as a baseline.

Given the very imminent US elections, the example used throughout the paper should probably be updated.

---

> ### Author Response · Authors · 2024-11-16
>
> We greatly appreciate your constructive feedback and the time you took to review our work. We will carefully address each of your points raised.
>
> ### Weaknesses
>
> **An FT Baseline:**
> - Apologies for any confusion caused! Including such a baseline is indeed important. In fact, FT-M and FT-L serve as the FT baselines for this purpose. To ensure comparability with our method, we updated the same layers as MEMIT for FT-M and FT-L (see Section D.2) and evaluated them on all datasets rather than directly copying numbers from the original survey.
> - We have clarified this in the updated manuscript and moved this discussion to the main text.
>
> **Clarifications for Section 3.2:**
> - This section was intended to provide readers with insights that when no external information is provided, there is no benefit from sampling, leading to the final method. We have updated the manuscript to make this point clearer.
>
> **Novelty of the Method:**
> - Regarding Snell et al. (2022), we agree that conceptually, both methods aim to learn context information efficiently and effectively. However, they diverge in their algorithmic and technical approaches. Snell et al. (2022) requires two copies of model weights because it is fundamentally a distillation algorithm (although the weights may originate from the same model). In contrast, our method proposes a regularization term that can be seamlessly added during fine-tuning. This makes our method easier to apply and tune via the \(\lambda\) parameter.
> - Additionally, Snell et al. (2022) relies on task templates and answer extraction modules. Overall, our method can be more easily applied to learning problems. We have updated the manuscript to discuss these distinctions in the related work section.
>
> ### Questions
>
> **$\lambda$ Used in the Experiments:**
> - In the experiments, we used $\lambda = 1.0$ without deliberate tuning. During the rebuttal period, we conducted ablation studies on $\lambda$ and have included these results in the updated manuscript.
>
> **Portability Score of the WikiBio Dataset:**
> - The WikiBio dataset does not include data for hopping questions, which is why no portability score was evaluated for it.
>
> **Perplexity Score:**
> - Thank you for raising this question! The perplexity score measures how well the pre-trained model predicts the generated outputs from the fine-tuned models. Assuming the original model is well-trained, the perplexity score reflects the quality of the fine-tuned model and how far it has drifted. We have clarified this in the updated manuscript.
>
> **Best Model Not Reported as a Baseline:**
> - Apologies for the confusion. Our statement referred to FT-M demonstrating nearly the best performance according to the survey, and FT-M is indeed included as a baseline in our experiments.
>
> **Updated Layers:**
> - Thank you for highlighting this! These details are essential, and we have updated the main text to make sure they are clear.
>
> We sincerely appreciate your valuable feedback and would be grateful for any further thoughts you may have!

---

> ### Author Response · Authors · 2024-11-24
>
> Dear Reviewer c9Gw,
>
> We sincerely appreciate your time and effort in reviewing our work. We understand that your schedule may be quite busy. As the authors-reviewer discussion phase draws to a close, we kindly request your attention to our responses. Our aim is to gain insights into whether our responses effectively address your concerns and to ascertain if there are any additional questions or points you would like to discuss. We also hope that if you are satisfied with our answers, you may consider adjusting your score and confidence accordingly.
>
> We look forward to the opportunity to discuss this further with you. Thank you for your thoughtful consideration.

---

> > ### Comment · Reviewer_c9Gw · 2024-11-26
> >
> > Thank you for the response.
> >
> > It seems the missing baseline is actually included, it was just not clear from the presentation. I have increased my score accordingly. Although it is still a bit unclear which of the two baselines is then the lambda=0 equivalent.
> >
> > Regarding novelty compared to Snell et al (2022): Even though you frame this conceptually as a regulariser here, the practical method is still essentially the same. The author response did not present any specific differences. This previous work should be highlighted accordingly in the paper.

---

> > > ### Author Response · Authors · 2024-11-26
> > >
> > > Dear Reviewer c9Gw,
> > >
> > > Thank you for your thoughtful comments and for updating your score. We appreciate your point regarding Snell et al. (2022) and have revised the related work section to address it accordingly. Your feedback has been invaluable in helping us improve our work.

---

### Official Review · Reviewer_2ELD · 2024-11-02

**Soundness:** 3
**Presentation:** 3
**Contribution:** 3
**Rating:** 8
**Confidence:** 3

**Summary:**

This paper proposed a method for knowledge editing that is based on leveraging the in-context ability of the model. To learn a new fact by finetuning, the methods proposes to train the model with supervision coming from itself when it can answer the query correctly based on contextual information.

What is interesting is that since the method relies and benefits from the in-context adapting ability of the model to have better update, better models would benefit even more from finetuning with this method [1, 2].

Overall, this is a good contribution with interesting potential for future work. I think the author could include a discussion about how much the base in-context ability of the models impact the success of the method.

[1] Yu et al 2023 Characterizing mechanisms for factual recall in language models.
[2] Monea et al. 2024. A Glitch in the Matrix? Locating and Detecting Language Model Grounding with Fakepedia

**Strengths:**

The paper proposes an interesting methods that is likely to be useful for future applications.
The paper does a good job at demonstrating the usefulness of the method.

**Weaknesses:**

The paper only studies one base model, it is not clear how this generalizes to other model. In particular, I suspect that the size of the model and their base in-context capabilities might play an important role in the success of the method.

**Questions:**

I am curious to hear the thoughts of the authors about how much the base in-context ability of the models impact the success of the method.
The discussion section could be expanded with such consideration. It would be interesting especially since the paper does not experiment with different base models

---

> ### Author Response · Authors · 2024-11-16
>
> We are truly grateful for your encouraging remarks, insightful feedback, and thoughtful questions! We would like to address each point you have raised.
>
> ### Weaknesses
> **Generalization to Other Models:**
> - We apologize for any confusion! We have also conducted evaluations using GPT2, with results presented in Appendix Section D.3. Our findings reveal that GPT2 performs worse than Llama2, underscoring the importance of the base model's in-context capabilities.
>
> ### Questions
> **Impact of In-Context Ability:**
> - We agree that the effectiveness of ICE may be affected by the in-context abilities of the base model. This hypothesis is elaborated upon in the last two assumptions in Section 3.5 and is corroborated by the comparative results between GPT2 and Llama2.

---

> > ### Comment · Reviewer_2ELD · 2024-11-25
> > **Thanks for the answer**
> >
> > Thank you for the clarification and answers, I'll keep my 'accept' score

---

> > > ### Author Response · Authors · 2024-11-26
> > >
> > > Dear Reviewer 2ELD,
> > >
> > > Thank you for your support and thoughtful review, we sincerely appreciate your invaluable comments and feedback throughout this process.

---

### Official Review · Reviewer_Ectm · 2024-11-04

**Soundness:** 3
**Presentation:** 4
**Contribution:** 3
**Rating:** 6
**Confidence:** 5

**Summary:**

This work presents an auxiliary loss for finetuning an LLM which steers it towards new knowledge and applied in the knowledge editing domain. This is through “in-context editing” (ICE), which minimizes the distances of the output *distribution* of the original model without the new knowledge to that of a model which is conditioned on the new knowledge in the prompt. Besides accuracy and fluency, other facets of generation are evaluated, like whether unrelated knowledge is affected and whether the learned new knowledge generalizes to related knowledge. Compared to prior methods (ROME, MEMIT, FT-L, FT-M), ICE performs relatively well on most metrics.

**Strengths:**

1. The method is novel for knowledge editing by identifying an issue with prior approaches for targeted knowledge editing. While there has been prior work on fine-tuning and naive work on in-context (prompt-based) knowledge editing, the combination of distilling the in-context editing directly into the parameters has not been done.

2. The empirical results, while not perfect on all metrics and datasets, show promise across the baseline methods presented and on the standard metrics and perplexity.

3. Ablation studies emphasize the importance of both dynamically updating the target distribution during training and on the importance of the context. In particular, an in-depth analysis of the static target distribution shows that dynamic targets actually lead to better convergence.

4. More analysis shows that as the model is edited (updated) more, the degradation of the model is less prominent than the other baseline methods.

**Weaknesses:**

1. The method is similar to knowledge/context distillation or gisting, and so a connection should be drawn there. Still, applying this method appears novel for knowledge editing. However the lack of references to KD/gisting makes it hard to place how related (or not) this idea is to that line of work.

[Snell et al., 2022](https://arxiv.org/abs/2209.15189) - Context Distillation

[Mu et al., 2023](https://arxiv.org/abs/2304.08467) - Gisting

2. The paper advocates for conditioning on “context” to generate target token distributions. It isn’t clear based on the data that this “context” is what makes the ICE method good, as opposed to the training objective.

  The method requires GPT-4 outputs to generate the context, while the other baselines being compared against are not allowed any access to this (external model) context. If the method is instead interpreted as knowledge distillation, it is less clear how whether this is possible without this context from a stronger LLM.

  Concretely, the effect of context should be isolated from the distillation-like training objective. The latter targets the one-hot problem motivated by the introduction of the paper, but the former is discussed heavily by this paper.

  If I understand the ablations table correctly, the rows with x in “Context” gives us a view of what the distillation-like objective could do for model editing. It looks competitive (or even better) than the baselines presented in Table 3, and so I wonder what the context actually adds?

3. In addition, one way to explore this would be to measure an “upper bound” of how good the model could be if it were perfect with context, i.e. evaluate the model corresponding to $p_\theta(x, | c, q)$ both before and after training.

4. The baselines included do not seem comprehensive and are a little misleading. In particular, MEND and SERAC are other method mentioned by Zhang et al., 2024 [survey] that achieve strong results only slightly worse than FT-M. The omission of those results make it seem like ICE is tied with FT-M, both much better than other methods. In reality, ROME/MEMIT are relatively weak baselines compared to the other methods.
5. Another limitation of the method that should be acknowledged or perhaps even addressed directly is the number of modified parameters and cost to make the edits. MEMIT/ROME are local, while FT-M, FT-L, and ICE are full-model. But my understanding is that the latter 3 are actually similar in terms of training cost and modified parameters

[Mitchell et al., 2021](https://arxiv.org/abs/2110.11309) - MEND

[Mitchell et al., 2022](https://proceedings.mlr.press/v162/mitchell22a/mitchell22a.pdf) - SERAC.

**Questions:**

Questions


1. I don’t understand the direct optimization problem stated in L238 - why can't the loss be backpropped jointly through both distributions? Anyway, the solution of freezing the target distribution makes sense, but then I'm confused by the $s$ vs. $s+1$ in Equation 7 – wouldn't it be more accurate that both subscripts should be $\theta_{s}$ except we do not backprop through the left term of the KL term? That is what is stated (and drawn in the figure) but having $s+1$ in the subscript feels wrong because that’s in the future?

2. How are stop/end-of-sequence tokens controlled in the output sequence, and how is fluency measured with respect to that? In the examples shown in D.6, all of the methods start to generate more (possibly unrelated) text after giving the correct answer. How does that unrelated text get factored into the various metrics?

3. I’m confused by “Observation 1” and the subsequent proof because empirically, this is the same as row 4 in the ablations table 4, and it looks like it is substantially better than the fine-tuning baselines. Actually, I’m confused whether 3.1 and 3.2 are actually represented empirically anywhere so we have a sense of where it stands relative to other methods.

4. This was mentioned above, but to ask more directly mostly out of curiosity: what is the cost (in terms of training time + prompting) for this loss? Is it substantially slower or faster than FT-M, and what about after factoring any hyperparameter tuning?

## Other minor comments not affecting the judgement:

In the implementation details, GPT-2 is mentioned but it isn’t mentioned elsewhere in the paper. It is in the appendix, and so the mention of GPT-2 would be less confusing if it was moved entirely to the appendix.

Throughout the paper, `` should be used instead of '' to start quotations.

I'd also suggest renaming the abbreviation ICE to CICE, as ICE was already used to refer to in-context editing in [Cohen et al., 2023](https://arxiv.org/pdf/2307.12976), which is also cited in the paper as [6].

Finally, contemporary work worth knowing about and citing in a future draft: [Rozner et al., 2024](https://arxiv.org/abs/2406.09920).

---

> ### Author Response · Authors · 2024-11-16
>
> We sincerely appreicate your detailed and valuable feedback! Below we address each points acoordingly:
>
> **Weaknesses:**
>
> **Connections to Knowledge Distillation (KD) and Gisting:**
> We appreciate the suggestion! We absolutely agree that conceptually, all three methods aim to learn context information efficiently and effectively. However, they diverge in their algorithmic and technical approaches.
> - Snell et al. 2022 requires two copies of model weights because it is fundamentally a distillation algorithm (although the weights may originate from the same model). In contrast, our method proposes a regularization term that can be seamlessly added during fine-tuning. This makes our method easier to apply and tune via the \(\lambda\) parameter.
> - Mu et al. 2023 proposes a gisting technique that compresses the information of contexts into a token. This requires the addition of an extra token to the input, which may not be desirable in some cases, especially when we do not know that the context is useful beforehand.
>
> In summary, our method offers ease of application for learning and subsequent inference. We have revised the manuscript to discuss these distinctions in the related work section.
>
> **Effect of Context:**
> To clarify, the context refers to **the knowledge we aim to edit** or its paraphrase. The key point is that sampling without external information fails to produce an informative distribution; thus, incorporating knowledge as context is essential for effective tuning.
>
> Consequently, it is not critical whether the knowledge originates from a more powerful LLM. What matters is that the training objective is contingent upon the context. Without this context, the process would effectively mirror standard fine-tuning.
>
> To provide a clearer understanding of what the contexts entail, here are some examples:
> ```json
> {
>     "prompt": "The mother of Mallory Reaves is whom?",
>     "ground_truth": [
>         "Brynne Chandler"
>     ],
>     "target_new": "Lalli Reaves",
>     "context": [
>         "Mallory Reaves's mother is Lalli Reaves.",
>         "Lalli Reaves is the mother of Mallory Reaves.",
>         "The mother of Mallory Reaves is identified as Lalli Reaves.",
>         "In terms of parentage, Mallory Reaves's mother is Lalli Reaves.",
>         "Mallory Reaves is the child of Lalli Reaves."
>     ]
> }
> ```
>
> **Ablation Studies:**
> We apologize for any confusion caused! In our ablation studies, all rows are obtained with our combined loss function with sampling sequences. Specifically, 1) “Dynamic” refers to whether we consistently utilize the **unedited** model for sequence generation during the optimization process, and 2) "Context" indicates whether context is employed in the sampling process. Consequently, the four rows correspond to:
> 1. ICE
> 2. The method described in Section 3.2, which samples without context but updates the distribution during optimization, effectively equivalent to vanilla fine-tuning.
> 3. Samples generated with context from the unedited model, without updating the distribution during optimization.
> 4. Samples generated without context from the unedited model, and without updating the distribution during optimization.
>
> - **Ablations on Context**
> Rows 2 and 4 both sample sequences without context. Notably, row 4 can be competitive in terms of locality, primarily not because of the missing context but because of the use of samples from the unedited model, which better preserves the original weights. In contrast, row 2 performs worse than other baselines.
> - **Empirical Results for Section 3.1 and 3.2**
> The second row presents results for Section 3.2, which is equivalent to vanilla fine-tuning as indicated by Observation 1. This variation underperforms compared to other configurations.
>
>
> **Upper Bound Experiments:**
> - Thank you for the insightful suggestion! There is existing research that advocates for the use of in-context learning for knowledge editing [Zheng et al](https://arxiv.org/abs/2305.12740), which incorporates the edited knowledge directly as context. This approach achieves near-perfect accuracy and demonstrates greater locality than what ICE currently offers. However, it is important to note that this does not represent the absolute upper bound of the method, as additional techniques can be integrated to further enhance performance.
>
>
> **Additional Baselines (MEND and SERAC):**
> - Thank you! We will incorporate MEND and SERAC as additional baselines in our revised experiments to provide a more comprehensive comparison.
>
> **Modified Parameters and Cost Analysis:**
> - In our experiments, we ensure fair comparisons by modifying the same five layers as MEMIT for FT-M, FT-L, and ICE. The training costs for these three approaches are comparable. However, we recognize that the actual running time for ICE is slightly longer than for FT due to the need for sampling during training. Nonetheless, this difference is minor, as inference can be performed quickly compared to tuning.

---

> > ### Author Response · Authors · 2024-11-16
> >
> > **Questions:**
> >
> > **Optimization Details (Eq. 7):**
> > - Directly propagating the loss through both distributions is not desirable, as we aim for a uni-directional optimization: we do not intend to draw $p_{\theta}(x | [c, q])$ towards $p_{\theta}(x | q)$. We have updated the manuscript to clarify this point.
> > - Regarding the subscripts, we apologize for the confusion caused. At the optimization step $s$, the model is parameterized by $\theta_{s}$, and our objective is to identify the optimal $\theta_{s+1}$, which is why Equation 7 is structured as it is.
> >
> > **End-of-Sequence Tokens and Fluency Metrics:**
> > - In the evaluation phase, we set the max_out_len to 100 as a standard practice. Fluency is assessed using 2- and 3-gram entropy terms. We discussed the shortcoming of this metric in the paper and proposed utilizing perplexity to evaluate the linguistic quality of the results. However, you are right that both metrics do not adequately capture the semantics of the sentences. Future works can consider more sophisticated metrics.
> >
> > **Minor Comments:**
> > - We have updated the main text to clarify that the results for GPT-2 are provided in the appendix.
> > - Rozner et al. (2024) has been referenced in the related work section.
> > - The quotation signs have been updated.
> >
> >
> > Again, we sincerely appreciate your invaluable feedback, which has significantly enhanced our manuscript.

---

> > > ### Comment · Reviewer_Ectm · 2024-11-22
> > > **Response to comments by Authors**
> > >
> > > Thank you for explaining this in detail. I will update my subscore; however, at this time I don't think these answers change my overall recommendation - my understanding of the evaluation results is that ICE offers an interesting alternative approach for knowledge editing (compared to FT-M, MEND, SERAC) but does not do substantially better -- except in perplexity which is a metric added by this paper.
> > >
> > > A couple other minor comments/follow-up questions:
> > > I am still not convinced that maintaining 2 copies of the model weights is a fair characterization of the difference between this work and Snell 2022 - rather the ablations table is a better characterization. One main difference (as pointed out by c9Gw) is that you are generating/augmenting the context and so it is for a specific useful application setting. The other main architectural difference is that you are using a dynamic "teacher" model, which I think is different too. If you used a static model, then it would not be any different from needing 2 copies of the model, like Snell 2022. In other words, the bottom two rows of the Ablation tables would (I think) be closest to Snell 2022 (although of course not entirely the identical for several reasons).
> > >
> > > Back to Eq7, I think the notation is still confusing somehow - e.g on very first step ($s=1$), we would need to have $p_{\theta_2}(x\mid q)$ on the right side, which the model of a future step (i.e. we cannot compute that distribution since we don't have that model yet). Or we could interpret it as on the first step $s=1$, we are using the model $p_{\theta_0}$ on the left side and $p_{\theta_1}$ on the right. But that means we need to have a model from before the current iterations, which makes it sound like we are storing a copy of the model from a previous iteration in addition to the current one. The point you want to make is that $p_{\theta}$ is the same model parameters for both sides - except the left side is frozen (`requires_grad=False`). So maybe it would be clearer to define new notation like $p_{\theta^*}$ or even $p_{\theta^{\text{fixed}}}$ as a fixed distribution for the left side of the KL.
> > >
> > > For the output/EOS tokens, I'm not sure I understand yet - is it that there's no meaningful EOS token learned during finetuning, and so unless we build heuristics or use sentence segmentation, we would not know where to cut off the response?

---

> > > > ### Author Response · Authors · 2024-11-24
> > > >
> > > > We sincerely appreciate your detailed follow-up and thoughtful observations. Below, we address your points:
> > > >
> > > > 1.	Fair Characterization:
> > > > We agree that the ablations table effectively highlights the differences. Notably, our dynamic teacher model distinguishes ICE by enabling the progressive refinement of the target distribution during training, which is a key feature.
> > > > 2.	Clarification on Equation 7:
> > > > Thank you for pointing this out. To provide greater clarity, we have revised the notation to explicitly state:
> > > > $$
> > > > \theta_{s+1}^{*} = \arg\min_{\theta_{s+1}} L_{\text{ICE}} = \arg\min_{\theta_{s+1}} D_{\text{KL}} ( p_{\theta_s}(x | [c, q]) || p_{\theta_{s+1}}(x | q))
> > > > $$
> > > > We have updated the manuscript for improved readability.
> > > > 3.	End-of-Sequence Tokens and Fluency Metrics:
> > > > You are correct that fine-tuning does not explicitly optimize for EOS prediction. In our implementation, responses are truncated at a maximum length of 100 tokens. We acknowledge this as a limitation and appreciate your observation.
> > > >
> > > > We hope this addresses your concerns comprehensively. Thank you for your continued feedback—it is invaluable in refining our work!

---

> > > > > ### Comment · Reviewer_Ectm · 2024-11-25
> > > > >
> > > > > Thank you for the response, and I think I'm still confused by the statement of Eqn 7, but I think this is the kind of detail is not that important in the main paper and you should write whichever you feel is the clearest (because the other reviewers didn't have trouble with the original phrasing). The detailed algorithm is in Alg 1. anyway, for readers who want to know full details.
> > > > >
> > > > > Also, I think Fig 1(c) should be updated to match Eqn 7?
> > > > >
> > > > > In the end, I will keep my score unchanged (for reasons mentioned in my last message) but I think the current draft, along with clarifications after discussion, is a stronger paper than before, and I am more confident in the score I've given.

---

> > > > > > ### Author Response · Authors · 2024-11-26
> > > > > >
> > > > > > Dear Reviewer Ectm,
> > > > > >
> > > > > > Thank you for your detailed feedback and for updating your confidence level. Your insights have been invaluable in enhancing our manuscript.

---

### Official Review · Reviewer_yiJu · 2024-11-04

**Soundness:** 3
**Presentation:** 3
**Contribution:** 3
**Rating:** 8
**Confidence:** 4

**Summary:**

This paper presents ICE, a regularization loss that aims at addressing the limitations of the traditional fine-tuning loss to update knowledge. Experiments on the KnowEdit dataset show its effectiveness to update the model's knowledge especially in the continual editing setting compared to other baselines.

**Strengths:**

- The paper is clear, well-motivated and the idea is novel as far as I know.
- Compared to other baselines, this method is the only one capable of effectively editing knowledge continually.

**Weaknesses:**

- The pipeline is quite heavy, relying on sampling at every optimization step and GPT-4 for augmented contexts.

**Questions:**

- "we sample sequences x_c from the model conditioned on $[c, q, x^∗]$"(L.246): Could you explain the rationale behind including $x^*$ in the sampling of $x_s$ ? what happens if the sampling is only conditioned on $[c, q]$ without the target ? (which better corresponds to Eq.4).
- I'm concerned about the context generation process, you mention that you use GPT-4 however, it somehow defeats the purpose since GPT-4 can be prone to hallucinations due to its training data becoming obsolete. I'm really not convinced by the prompt that you used: *"Please help me generate five complete statements as [context]s according to the semantics of incomplete facts '{prompt}' and '{target}'."*. In fact, if GPT-4 hallucinates, it will provided non factual contexts, hindering the optimization process and potentially incurring further hallucinations. Consider adding a 'Limitations' section that addresses the potential risks of using GPT-4 for context generation, including the possibility of hallucinations or outdated information. This section could also explore potential mitigation strategies or alternative approaches for context generation.
- From L.431, could you provide a clear definition of what 'temperature' refers to in this context ?
- In Algorithm 1, only $L_\text{ICE}$ is shown in the optimization process. If $L_\text{FT}$ is also used, could you update the algorithm to reflect this? Additionally, an ablation study on the effect of $\lambda$ would provide valuable insights into the relative contributions of $L_\text{ICE}$ and $L_\text{FT}$ to the overall performance.

*Typos:*
- Figure 1: for the fine-tuning part, it should be $p_{\theta_s}(x | q)$ in the FT loss (and not $p_{\theta_s}(x)$).

---

> ### Author Response · Authors · 2024-11-16
>
> We sincerely appreciate your insightful feedback and questions! We value your perspectives and would like to address each point raised:
>
> ### Weaknesses
> Concerning the computational demands of our pipeline, it is accurate that our method involves multiple sampling steps and depends on GPT-4 for context generation. However, we would like to clarify that the computational burden may not be as substantial as it appears:
> 1) Since sampling only necessitates a forward pass of the model, the computational cost is significantly lower than that of training the model.
> 2) We are considering scenarios with very limited training data, as is the case in the knowledge editing task. Therefore, the computational cost is not as critical as it would be in large-scale pre-training.
>
> To get a more concrete idea of how the contexts look like, here are some examples:
> ```json
> {
>     "prompt": "The mother of Mallory Reaves is whom?",
>     "ground_truth": [
>         "Brynne Chandler"
>     ],
>     "target_new": "Lalli Reaves",
>     "context": [
>         "Mallory Reaves's mother is Lalli Reaves.",
>         "Lalli Reaves is the mother of Mallory Reaves.",
>         "The mother of Mallory Reaves is identified as Lalli Reaves.",
>         "In terms of parentage, Mallory Reaves's mother is Lalli Reaves.",
>         "Mallory Reaves is the child of Lalli Reaves."
>     ]
> }
> ```
> We have also updated the manuscript's appendix to include the examples.
>
> ### Questions
> **Rationale behind including $x^{\*}$ **
> - That is an excellent question! If the sampling is not conditioned on the target, we would be optimizing L_{ICE} alone. However, the inclusion of $x^{\*}$ allows us to optimize the combined loss $L_{ICE} + \lambda L_{FT}$. By directly maximizing the likelihood of $[x^{\*}, x_c]$, we can optimize the combined loss given $\lambda = 1$: $L = L_{ICE} + L_{FT}$, which is algorithmically convenient. We have included a proof of this in the updated appendix A.3.
>
> **Concerns regarding GPT-4 for context generation**
> - We understand your concerns regarding the use of GPT-4 for context generation. We recognize that GPT-4 is not infallible and can produce hallucinations, and we have added a 'Limitations' section in our revised manuscript that discusses the potential risks associated with using GPT-4. In practice, since we are utilizing the model solely for paraphrasing, we have not observed any hallucinations.
>
> **Definition of 'temperature'**:
> - In this context, the term "temperature" refers to the standard temperature sampling parameter used when sampling sequences from the model. The purpose here was to investigate how temperature influences optimization. While both optimization schemes demonstrate convergence, our method consistently exhibits lower equilibrium loss under different temperatures.
>
> **Updates to Algorithm 1**
> - Thank you for your suggestion! We have updated Algorithm 1 in the manuscript to reflect that. Additionally, we added an ablation study on the effect of $\lambda$ in the updated manuscript. In our original experiments, we simply used $\lambda=1$, which yielded satisfactory performance, and we did not deliberately tune this parameter, as it serves more as a tuning variable for others to experiment with.
>
> Thank you once again for your invaluable feedback!

---

> > ### Author Response · Authors · 2024-11-24
> >
> > Dear Reviewer yiJu,
> >
> > We sincerely appreciate your time and effort in reviewing our work. We understand that your schedule may be quite busy. As the authors-reviewer discussion phase draws to a close, we kindly request your attention to our responses. Our aim is to gain insights into whether our responses effectively address your concerns and to ascertain if there are any additional questions or points you would like to discuss. We also hope that if you are satisfied with our answers, you may consider adjusting your score and confidence accordingly.
> >
> > We look forward to the opportunity to discuss this further with you. Thank you for your thoughtful consideration.

---

> > ### Comment · Reviewer_yiJu · 2024-11-25
> >
> > I appreciate your efforts in revising the manuscript. I have updated my score accordingly.

---

> > > ### Author Response · Authors · 2024-11-26
> > >
> > > Dear Reviewer yiJu,
> > >
> > > Thank you for your thoughtful comments and for updating your score. Your feedback has been invaluable in improving our manuscript.

---

### Meta-Review · Area_Chair_PQDB · 2024-12-21

**Metareview:**

This paper introduces an optimization framework for knowledge editing in LLMs. Rather than optimize towards one-hot target distributions, the procedure instead optimizes the model towards a soft distribution induced by conditioning the source model on additional context.  This is combined with the standard optimization towards the one-hot target.

Compared to ROME/MEMIT and more basic fine-tuning, this approach has higher locality and lower perplexity (reflecting better preserved language quality) The approach is also effective in a continual editing setting.

This paper's method is an interesting approach for knowledge editing. It is well-written and the experimental evaluation is thorough.

There are a few weaknesses of the paper. One weakness raised by Ectm is the weakness of the baselines. The knowledge editing methods here are dated.  Furthermore, the improvements over the baselines are most substantial on perplexity, which is a metric introduced by this paper to measure fluency of the model.  A second weakness is that the methodological contribution of the paper may not be that general. The method is similar to other approaches, namely the work of Snell et al. (2022), so these ideas of distillation are already being used for related problems; this paper's contribution is chiefly to show that they work here as well.  A final weakness mentioned by the reviewers is the computational cost of the method, but this is not that substantial: the method is slower but not terribly so.

The reviewers end up positive about the paper, but there are a few reservations due to the weaknesses above. Notably, two reviewers (Ectm and c9Gw) remain a bit unimpressed by the novelty and experimental comparisons with the baselines.

Another paper that was raised in private discussion is this one https://arxiv.org/abs/2306.09306 which the authors could consider discussing.

**Additional Comments On Reviewer Discussion:**

yiJu brings up the computational cost, which in my view is not a major issue, and the authors address this.

Ectm mentions related prior work and asks a conceptual question about disentangling the role of context from the KL objective.  However, the biggest issue raised by this reviewer is the weak baselines. I agree: I think that comparing to just ROME, MEMIT, and fine-tuning is not sufficient, particularly when there are approaches like context distillation and gisting that are methodologically similar to this as well.

This point is not really resolved:

> Thank you for explaining this in detail. I will update my subscore; however, at this time I don't think these answers change my overall recommendation - my understanding of the evaluation results is that ICE offers an interesting alternative approach for knowledge editing (compared to FT-M, MEND, SERAC) but does not do substantially better -- except in perplexity which is a metric added by this paper.

c9Gw asked for an additional baseline, which was already included, and also mentioned the conceptual similarity to Snell et al.

---

### Decision · Program_Chairs · 2025-01-22

Accept (Poster)